# Audio-Driven Co-Speech Gesture Video Generation

**Xian Liu[1], Qianyi Wu[2], Hang Zhou[1], Yuanqi Du[3], Wayne Wu[4], Dahua Lin[1,4], Ziwei Liu[5]**
[1]Multimedia Laboratory, The Chinese University of Hong Kong   [2]Monash University
[3]Cornell University   [4]Shanghai AI Laboratory   [5]S-Lab, Nanyang Technological University

## Abstract

Co-speech gesture is crucial for human-machine interaction and digital entertainment. While previous works mostly map speech audio to human skeletons (*e.g.,* 2D keypoints), directly generating speakers' gestures in the image domain remains unsolved. In this work, we formally define and study this challenging problem of *audio-driven co-speech gesture video generation*, *i.e.,* using a *unified* framework to generate speaker *image sequence* driven by speech audio. Our key insight is that the co-speech gestures can be decomposed into common motion patterns and subtle rhythmic dynamics. To this end, we propose a novel framework, **A**udio-drive**N G**esture v**I**deo g**E**neration (**ANGIE**), to effectively capture the reusable co-speech gesture patterns as well as fine-grained rhythmic movements. To achieve high-fidelity image sequence generation, we leverage an unsupervised motion representation instead of a structural human body prior (*e.g.,* 2D skeletons). Specifically, **1)** we propose a vector quantized motion extractor (**VQ-Motion Extractor**) to summarize common co-speech gesture patterns from implicit motion representation to codebooks. **2)** Moreover, a co-speech gesture GPT with motion refinement (**Co-Speech GPT**) is devised to complement the subtle prosodic motion details. Extensive experiments demonstrate that our framework renders realistic and vivid co-speech gesture video. Demo video and more resources can be found in: https://alvinliu0.github.io/projects/ANGIE

## 1 Introduction

During daily conversation among humans, speakers naturally emit co-speech gestures to complement the verbal channels and express their thoughts [17, 35, 56]. Such non-verbal behaviors ease speech comprehension [10, 58] and bridge the communicator's gap for better credibility [7, 54]. Therefore, equipping the social robot with conversation skills constitutes a crucial step to human-machine interaction. To achieve it, researchers delve into the task of co-speech gesture generation [21, 39, 62], where audio-coherent human gesture sequences are synthesized in the form of structural human representation (*e.g.,* skeletons). However, such representation contains no appearance information of the target speaker, which is crucial for human perception. As demonstrated in audio-driven talking head synthesis [34, 65], generating real-world subjects in the image domain is highly desirable. To this end, we explore the problem of *audio-driven co-speech gesture video generation*, *i.e.,* using a *unified* framework to generate speaker *image sequence* driven by speech audio (illustrated in Fig. 1).

Conventional methods require exhaustive human efforts to pre-define the speech-gesture pairs and connection rules for coherent result [11, 12]. With the development of deep learning, neural networks are leveraged to learn the mapping from encoded audio feature to human skeletons in a data-driven manner [21, 39, 62]. Notably, one category of approaches relies on small-scale MoCap datasets in co-speech setting [16, 18, 48], which contributes to specific models with limited capacity and robustness. To capture more general speech-gesture correlations, another category of methods builds large training corpus by exploiting off-the-shelf pose estimators [9, 15] to label enormous online videos as pseudo ground truth [21, 63]. However, the inaccurate pose annotations induce error

36th Conference on Neural Information Processing Systems (NeurIPS 2022).

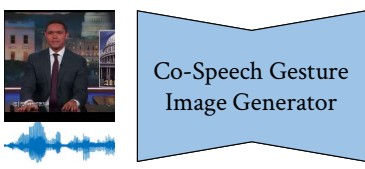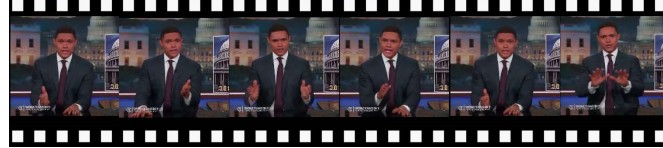

Figure 1: **Illustration of Problem Setting.** In this paper, we focus on audio-driven co-speech gesture video generation. Given an image with speech audio, we generate aligned speaker *image sequence*.

accumulation in the training phase, which makes the generated results unnatural. Besides, most previous works ignore the problem of co-speech gesture video generation. Only few works [21, 39] animate in the image domain as an *independent post-processing step*, which borrows from the existing pose-to-image generators [5, 13] to train on the target person's images. How to design a *unified* framework to generate speaker *image sequence* driven by speech audio remains unsolved.

To effectively learn the mapping from audio to co-speech gesture video, we pinpoint two important observations from current studies: 1) hand-crafted structural human priors like 2D/3D skeletons would eliminate articulated human body region information. Such a zeroth-order motion representation fails to formulate first-order motion like local affine transformation in image animation [44]. Besides, the error in structural prior labeling impairs cross-modal audio-to-gesture learning [33]. 2) Motivated by previous linguistic studies [27, 47], the co-speech gestures could be decomposed into common motion patterns and rhythmic dynamics, where the former ones refer to large-scale motion templates (*e.g.,* periodically put hands up and down), while the latter ones play a refinement role to complement subtle prosodic movements and synchronize with speech audio (*e.g.,* finger flickers).

We take inspiration from the above observations and propose a novel framework **A**udio-drive**N G**esture v**I**deo g**E**neration (**ANGIE**) to generate co-speech gesture video. The key insight is to *summarize common co-speech gesture patterns from motion representation to quantized codebooks* and further *refine subtle rhythmic details by motion residuals for fine-grained results*. In particular, two modules are designed, namely **VQ-Motion Extractor** and **Co-Speech GPT**. In VQ-Motion Extractor, we utilize an unsupervised motion representation to depict the articulated human body and first-order gestures [45]. The codebooks are established to quantize the reusable common co-speech gesture patterns from unsupervised motion representation. To guarantee the validity of gesture patterns, we propose a cholesky decomposition based quantization scheme to relax the motion component constraint. The position-irrelevant motion pattern is extracted as final quantization target to represent the relative motion. In this way, the quantized codebooks naturally contain rich common gesture pattern information. With the quantized motion code sequence, in Co-Speech GPT we use a GPT-like [40] structure to predict discrete motion patterns from speech audio. Finally, a motion refinement network is used to complement subtle rhythmic details for fine-grained results.

To summarize, our main contributions are three-fold: **1)** We explore a challenging problem of audio-driven co-speech gesture *video* generation. To the best of our knowledge, we are the first to generate co-speech gesture in image domain with a *unified* framework *without any structural human body prior*. **2)** We propose the VQ-Motion Extractor to quantize the motion representation into common gesture patterns and the Co-Speech GPT to refine subtle rhythmic movement details. The codebooks naturally contain reusable motion pattern information. **3)** Extensive experiments demonstrate that the proposed framework **ANGIE** renders realistic and vivid co-speech gesture video generation results.

## 2   Related Work

**Co-Speech Gesture Generation.** Synthesizing co-speech gesture has gained research interest in vision [3, 21, 29], graphics [4, 60, 62] and robotics [23, 25, 63] domains. Recent researches resort to deep neural networks to learn the speech-gesture mapping in a data-driven manner, with major focuses on below perspectives: 1) Dataset. One strand of methods use small-scale MoCap datasets to learn specific models [16, 18, 30, 42, 48, 51, 55], while another strand of works exploit off-the-shelf estimator to label enormous videos as structural prior [1–3, 21, 33, 39, 61–63]. The dataset scale *v.s.* pose annotation accuracy often acts as a trade-off in this task: A large amount of speech-gesture pairs facilitate the training of more general models with better capacity and robustness, yet error accumulation in annotations induces unnatural results. 2) Framework architecture. CNN-based [21],

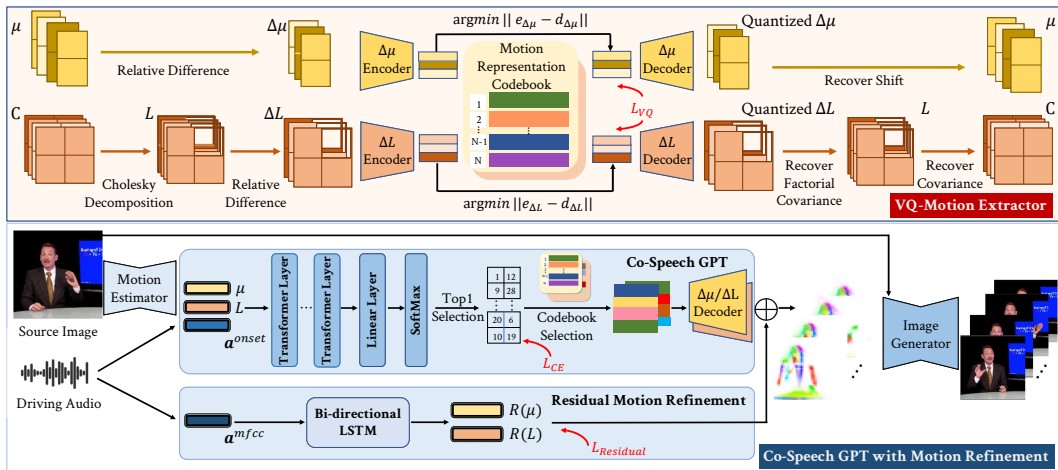

Figure 2: **Overview of the Audio-driveN Gesture vIdeo gEneration (ANGIE) framework**. In VQ-Motion Extractor, the cholesky decomposition with position-irrelevant design transforms the shift-translation $\mu$ and covariance $C$ to relative motion pattern representation of $\Delta\mu$ and $\Delta L$, which are further quantized by codebooks to extract the common motion patterns. Given the driving audio and starting gesture codes, the Co-Speech GPT predicts the future motion fields. A Motion Refinement network further learns motion residuals to complement the subtle rhythmic dynamics.

RNN-based [62] and Transformer-based [6] frameworks show promising results. To further improve the diversity of generated gestures and grasp the fine-grained cross-modal associations, components like adversarial loss [21], VAE sampling [30, 59] and hierarchical encoder-decoder design [33] are proposed. 3) Input modality. Some approaches treat single modality of speech audio [2, 19, 21–23, 30, 39] or text transcription [3, 6, 25, 63] as input to drive the co-speech gesture, while some others use both modalities as stimuli for generation [1, 33, 62]. To ease the learning of implicit cross-modal mapping from speech to gesture and create more stable results, recent works involve auxiliary input modality such as speaker style [2], pose mode [59] and motion template [39].

In this work, we take a step further in the above three aspects: 1) For the dataset, we collect a new co-speech gesture dataset in *image* domain, where an unsupervised motion representation is used to model articulated human body and bypass the inaccuracy from structural prior annotations. 2) For the architecture, a vector quantized (VQ) network with novel discretization scheme is proposed to extract valid relative motion patterns. We further devise a motion refinement network to complement subtle rhythmic dynamics. 3) For the input modality, we explicitly decouple the common motion pattern from co-speech gestures, which serves a similar role as motion template [39] to provide auxiliary input. However, our discrete codebook design is more suitable for finite gesture patterns than continuous representation, which is also proven in recent cross-modal generation tasks [36, 41, 46]. Furthermore, we propose a novel vector quantization network with cholesky decomposition scheme to extract the valid motion patterns. We improve the quantization scheme to encode the relative motion representation that is position (absolute location) irrelevant. A motion refinement network is further devised to complement subtle rhythmic dynamics. Notably, our approach gives an idea on how to deal with the constraints in vector quantization and how to complement sequential results with missing details. Such design could prospectively provide insights for relevant domains like constrained vector quantization problem, cross-modal learning [32] and video generation tasks [46].

**Video/Audio-Driven Video Generation.** Traditional video-driven approaches for image animation can be categorized into supervised and unsupervised, where the supervised methods typically involve structural human body prior such as landmarks [8, 64] and 3D parametric models [20, 50], while the unsupervised approaches design self-supervised tasks to animate unlabeled images [43–45, 57]. To facilitate broader applications, researchers explore audio-driven video generation, where one of the most relevant tasks is talking face generation [14, 38]. Different from the strong correlations between audio and mouth shape, the mapping from audio to complicated co-speech motion is multi-modal and harder to learn. Most co-speech gesture studies synthesize human skeletons as final results (*e.g.,* 2D keypoints), while only few works [21, 39] generate co-speech images in a *post-processing* manner.

# 3 Our Approach

We present **ANGIE** that generates audio-driven co-speech gesture in image domain, where the speakers' image sequence is driven by speech audio as shown in Fig. 1. The whole pipeline is illustrated in Fig. 2. To make the content self-contained and narration clearer, we first introduce the preliminaries and problem setting in Sec. 3.1. Then, we present the **VQ-Motion Extractor** which extracts common co-speech gesture patterns as quantized codebooks in Sec. 3.2. Finally, we elaborate the **Co-Speech GPT** to complement subtle rhythmic dynamics for fine-grained results in Sec. 3.3.

## 3.1 Preliminaries and Problem Setting

**Unsupervised Motion Representation.** To achieve high-fidelity image animation, we take inspiration from MRAA [45] that uses an unsupervised motion representation to drive articulated objects. MRAA first estimates a coarse motion representation from the source and driving frames, then predicts dense pixel-wise flow for image generation. Specifically, an encoder-decoder keypoint predictor produces $K$ different heatmaps $\mathbf{H}^1, \mathbf{H}^2, \ldots, \mathbf{H}^K$, where $K$ is region number and $\mathbf{H}^k$ denotes the $k$-th image region. Afterwards, each heatmap is normalized by softmax operation, *i.e*, $\sum_{z \in \mathcal{Z}} \mathbf{H}^k(z) = 1$, where $z$ is the image pixel location and $\mathcal{Z}$ is the set of all pixels. The key insight behind MRAA is to represent each region's motion by affine transformation with a shift-translation component. The shift-translation component $\mu^k \in \mathbb{R}^2$ and the distribution $C^k$ of the $k$-th part can be calculated as:

$$\mu^k = \sum_{z \in \mathcal{Z}} \mathbf{H}^k(z)z, \quad C^k = \sum_{z \in \mathcal{Z}} \mathbf{H}^k(z)(z - \mu^k)(z - \mu^k)^\mathsf{T}, \tag{1}$$

where $\mathsf{T}$ is matrix transpose, $C^k \in \mathbb{R}^{2 \times 2}$ measures the covariance of heatmap value. It naturally depicts the size and shape of an articulated region. To represent the affine transformation $A^k \in \mathbb{R}^{2 \times 2}$ of the $k$-th region, we apply singular value decomposition (SVD) to $C^k$ and derive $A^k$ as:

$$C^k = U^k \Sigma^k (V^k)^\mathsf{T}, \quad A^k = U^k \Sigma^{k \frac{1}{2}}, \tag{2}$$

where unitary matrices $U^k, V^k$ and diagonal matrix $\Sigma^k$ are the SVD result of covariance matrix $C^k$. The representation $\mathcal{M}$ extracted by motion estimator is the concatenation of $[\mu; C; A] \in \mathbb{R}^{K \times (2+4+4)}$ for $K$ distinct regions, an image generation module with dense pixel-wise flow predictor synthesizes the final generation results. In this work, the motion representation $\mathcal{M}$ and image generation module $G$ generally follow MRAA. We suggest the readers referring to [45] for more details.

**Problem Setting for Co-Speech Gesture Image Generation.** We collect training data of massively-available speaking videos with clear co-speech gestures for natural self-reconstruction supervision. Specifically, given an $(N + 1)$-frame video clip $\mathbf{V} = \{I_{(0)}, \ldots, I_{(N)}\}$, the goal of our framework at the training stage is to predict the motion representation $\widehat{\mathcal{M}}_{(1:N)}$ based on the first image frame $I_{(0)}$ and video's accompanying audio sequence $\mathbf{a} = \{a_{(1)}, \ldots, a_{(N)}\}$. Further, the image generation module $G$ reconstructs the video frames $\widehat{I}_{(1:N)}$. At the inference stage, an arbitrary reference image with speech audio clip is provided to generate subsequent image frames. According to the observation in Sec. 1, we decompose the co-speech gestures into common motion patterns and subtle rhythmic dynamics. The overall training setting can be formulated as:

$$\widehat{I}_{(1:N)} = G(I_{(0)}, \widehat{\mathcal{M}}_{(1:N)}(\mathbf{a})), \qquad \widehat{\mathcal{M}}_{(1:N)} = \widehat{\mathcal{M}}^{pattern}_{(1:N)} + \widehat{\mathcal{M}}^{rhythmic}_{(1:N)}, \tag{3}$$

where $\mathcal{M}^{pattern}$ denotes the gesture pattern (Sec. 3.2) and $\mathcal{M}^{rhythmic}$ is rhythmic movement (Sec. 3.3).

## 3.2 Vector Quantized Motion Pattern Extractor

To decompose the co-speech gestures, we propose to firstly extract the common motion patterns. However, three problems remain: 1) The gesture sequences are different from each other. While some motion sequences share the same action pattern, the dynamic details may vary a lot. How to extract the major motion pattern despite the influence from minor prosodic movements? 2) The covariance matrix $C$ is symmetric positive definite (Eq. 1), which further constrains the range of affine matrix $A$. How to preserve such characteristic for valid gesture patterns? 3) Since the unsupervised motion representation is extracted in the image pixel space, it is affected by the absolute location of each articulated region. How to represent the position-irrelevant motion pattern information?

**Vector Quantized Motion Pattern Learning.** Our solution to the first problem is to quantize the common motion pattern into a codebook. Since the gesture pattern is finite, it could be summarized to discrete codebook entries. Besides, each codebook entry refers to a certain type of gesture pattern, which matches our goal to extract the common and reusable co-speech gesture patterns.

A *naive* way is to quantize the motion representation $\mathcal{M}$ separately as shift-translation $\mu$, covariance matrix $C$ and affine transformation $A$. Specifically, for a $T$-frame co-speech gesture sequence $I_{(1:T)}$, we transform it into $[\mu; C; A]_{(1:T)} \in \mathbb{R}^{T \times K \times (2+4+4)}$, where $[\mu; C; A] = \mathcal{M}$ denotes motion representation of $K$ regions as in Sec. 3.1. We first build three codebooks $\mathcal{D}_\mu = \{\mathbf{d}_{\mu,m}\}_{m=1}^M$, $\mathcal{D}_C = \{\mathbf{d}_{C,m}\}_{m=1}^M$ and $\mathcal{D}_A = \{\mathbf{d}_{A,m}\}_{m=1}^M$ for each motion component respectively, where $M$ is codebook size. $\mathbf{d}_{\mu,m}, \mathbf{d}_{C,m}$ and $\mathbf{d}_{A,m} \in \mathbb{R}^\ell$ are the $m$-th entry of $\ell$-channel for each codebook. Then, three separate encoders $E_\mu$, $E_C$ and $E_A$ are utilized to encode the corresponding context information into latent features of $\mathbf{e}_\mu = \{\mathbf{e}_{\mu,i}\}_{i=1}^{T'}, \mathbf{e}_C = \{\mathbf{e}_{C,i}\}_{i=1}^{T'}$ and $\mathbf{e}_A = \{\mathbf{e}_{A,i}\}_{i=1}^{T'} \in \mathbb{R}^{T' \times \ell}$, where $T'$ is the temporal dimension and $\ell$ is the channel dimension. Notably, we denote the $i$-th temporal feature of each motion component as $\mathbf{e}_{\mu,i}$, $\mathbf{e}_{C,i}$ and $\mathbf{e}_{A,i}$. The feature encoding process can be formulated as:

$$E_\mu(\mu_{(1:T)}) = \mathbf{e}_\mu, \quad E_C(C_{(1:T)}) = \mathbf{e}_C, \quad E_A(A_{(1:T)}) = \mathbf{e}_A. \tag{4}$$

Following the pipeline of VQ-VAE [53], we individually quantize $\mathbf{e}_\mu$, $\mathbf{e}_C$ and $\mathbf{e}_A$ by substituting each temporal feature $\mathbf{e}_{\mu,i}$, $\mathbf{e}_{C,i}$ and $\mathbf{e}_{A,i}$ to the nearest codebook entry $\mathbf{d}_{\mu,m}$, $\mathbf{d}_{C,m}$ and $\mathbf{d}_{A,m}$ as:

$$\underbrace{\mathbf{e}_\mu^{\mathbf{q}} = \arg\min_{\mathbf{d}_\mu \in \mathcal{D}_\mu} ||\mathbf{e}_\mu - \mathbf{d}_\mu||,}_{\textit{quantize shift-translation } \mu} \quad \underbrace{\mathbf{e}_C^{\mathbf{q}} = \arg\min_{\mathbf{d}_C \in \mathcal{D}_C} ||\mathbf{e}_C - \mathbf{d}_C||,}_{\textit{quantize covariance matrix } C} \quad \underbrace{\mathbf{e}_A^{\mathbf{q}} = \arg\min_{\mathbf{d}_A \in \mathcal{D}_A} ||\mathbf{e}_A - \mathbf{d}_A||,}_{\textit{quantize affine transformation } A} \tag{5}$$

where $\mathbf{e}_\mu^{\mathbf{q}} = \{\mathbf{e}_{\mu,i}^{\mathbf{q}}\}_{i=1}^{T'}, \mathbf{e}_C^{\mathbf{q}} = \{\mathbf{e}_{C,i}^{\mathbf{q}}\}_{i=1}^{T'}$ and $\mathbf{e}_A^{\mathbf{q}} = \{\mathbf{e}_{A,i}^{\mathbf{q}}\}_{i=1}^{T'} \in \mathbb{R}^{T' \times \ell}$ are the quantized code sequence of length $T'$ for each motion component. The $i$-th quantized code of each motion component is denoted as $\mathbf{e}_{\mu,i}^{\mathbf{q}}, \mathbf{e}_{C,i}^{\mathbf{q}}$ and $\mathbf{e}_{A,i}^{\mathbf{q}}$, respectively. Finally, three separate decoders $D_\mu$, $D_C$ and $D_A$ are leveraged to reconstruct the motion representations of each component as:

$$\widehat{\mu}_{(1:T)} = D_\mu(\mathbf{e}_\mu^{\mathbf{q}}), \quad \widehat{C}_{(1:T)} = D_C(\mathbf{e}_C^{\mathbf{q}}), \quad \widehat{A}_{(1:T)} = D_A(\mathbf{e}_A^{\mathbf{q}}). \tag{6}$$

Such discrete representation also ease the audio-to-gesture learning (Sec. 3.3): Previous methods predict continuous output as a *harder regression* problem. While we only need to predict features nearer to the correct codebook entry, which in essence resembles an *easier classification* problem.

**Quantization Design for Valid Motion Representation.** To extract valid gesture patterns, we have to preserve certain characteristics of motion representation. Especially, the covariance matrix $C$ should be symmetric positive definite (Eq. 1), and the affine transformation $A$ is determined by $C$ through SVD (Eq. 2). Therefore, instead of naively quantize each component in Eq. 5, we propose to only quantize the shift-translation $\mu$ and covariance matrix $C$, while derive the affine transformation $A$ with SVD. The only constraint is to guarantee that the covariance matrix $C$ is symmetric positive definite. To satisfy such requirement, we use the *unique cholesky decomposition theorem* [52]:

**Theorem 1.** *For any real symmetric positive definite matrix $C \in \mathbb{S}_{++}^n$, there exists a unique lower triangular matrix $L$ with positive diagonal entries, such that $C = LL^{\mathsf{T}}$.* $\qquad\square$

In this way, we turn to quantize the lower triangular matrix $L = \begin{pmatrix} l_1 & 0 \\ l_2 & l_3 \end{pmatrix}$, where the constraint is much simpler as $l_1, l_3 > 0$. The updated quantization scheme with cholesky decomposition is:

$$\underbrace{\mathbf{e}_\mu^{\mathbf{q}} = \arg\min_{\mathbf{d}_\mu \in \mathcal{D}_\mu} ||\mathbf{e}_\mu - \mathbf{d}_\mu||,}_{\textit{quantize shift-translation } \mu} \quad \underbrace{\mathbf{e}_L^{\mathbf{q}} = \arg\min_{\mathbf{d}_L \in \mathcal{D}_L} ||\mathbf{e}_L - \mathbf{d}_L||,}_{\textit{quantize the lower triangular matrix } L} \tag{7}$$

where $\mathbf{e}_L$, $\mathbf{e}_L^{\mathbf{q}}$, $\mathcal{D}_L$ and $\mathbf{d}_L$ denote the encoded feature, quantized feature, codebook and codebook entry for factorial covariance $L$, respectively. A simple transformation of $l_{1,3} = \texttt{ReLU}(l_{1,3}) + \epsilon$ guarantees the diagonal entries to be positive, where $\epsilon$ is a small positive number. The motion component $C$ and $A$ can be further obtained by $LL^{\mathsf{T}}$ and SVD calculation, respectively.

**Position-Irrelevant Motion Pattern.** Another problem arises when we inspect the value of the motion representation: Since $\mathcal{M}$ is extracted in the image pixel space, the object location will affect

the element in $\mathcal{M}$. For example, if a person poses the same gesture at different image regions, the motion component differs yet the underlying motion pattern remains the same. Thus we focus on a image location invariant motion pattern representation. In particular, due to the linear additiveness, the relative shift-translation $\mu$ between adjacent frames can be represented as $\Delta\mu_j = \mu_j - \mu_{j-1}$, and the relative change of the lower triangular matrix is $\Delta L_j = L_j - L_{j-1}$ for any $j = 2, \ldots, N$. Note that with the uniqueness of cholesky decomposition, $(L + \Delta L)$ corresponds to the sole covariance matrix $C = (L + \Delta L)(L + \Delta L)^{\mathsf{T}}$, which further determines affine matrix $A$ by SVD. In this way, the term $\Delta L$ is sufficient to represent any relative affine transformation between two frames. We accordingly update the quantization scheme with position-irrelevant motion pattern representation as:

$$\underbrace{\mathbf{e}^{\mathbf{q}}_{\Delta\mu} = \arg\min_{\mathbf{d}_{\Delta\mu}\in\mathcal{D}_{\Delta\mu}} ||\mathbf{e}_{\Delta\mu} - \mathbf{d}_{\Delta\mu}||,}_{\textit{quantize relative shift-translation } \Delta\mu} \quad \underbrace{\mathbf{e}^{\mathbf{q}}_{\Delta L} = \arg\min_{\mathbf{d}_{\Delta L}\in\mathcal{D}_{\Delta L}} ||\mathbf{e}_{\Delta L} - \mathbf{d}_{\Delta L}||}_{\textit{quantize relative lower triangular matrix change } \Delta L} \ , \quad (8)$$

where $\mathbf{e}_{\{\Delta\mu,\Delta L\}}$, $\mathbf{e}^{\mathbf{q}}_{\{\Delta\mu,\Delta L\}}$, $\mathcal{D}_{\{\Delta\mu,\Delta L\}}$ and $\mathbf{d}_{\{\Delta\mu,\Delta L\}}$ are the encoded feature, quantized feature, codebook and entry for the relative shift-translation $\Delta\mu$ and factorial covariance $\Delta L$, respectively.

**Overall Quantized Motion Pattern Learning.** With the position-irrelevant motion pattern, the codebook naturally contains reusable common co-speech gesture patterns $\mathcal{M}^{pattern}$. The encoders $E_{\Delta\mu}$, $E_{\Delta L}$ and the decoders $D_{\Delta\mu}$, $D_{\Delta\mu}$ are jointly learned with the codebooks $\mathcal{D}_{\Delta\mu}$ and $\mathcal{D}_{\Delta L}$ via:

$$\mathcal{L}_{VQ} = ||\widehat{\Delta\mu} - \Delta\mu|| + ||\mathbf{sg}[\mathbf{e}_{\Delta\mu}] - \mathbf{e}^{\mathbf{q}}_{\Delta\mu}|| + \beta_1||\mathbf{e}_{\Delta\mu} - \mathbf{sg}[\mathbf{e}^{\mathbf{q}}_{\Delta\mu}]||+$$
$$||\widehat{\Delta L} - \Delta L|| + ||\mathbf{sg}[\mathbf{e}_{\Delta L}] - \mathbf{e}^{\mathbf{q}}_{\Delta L}|| + \beta_2||\mathbf{e}_{\Delta L} - \mathbf{sg}[\mathbf{e}^{\mathbf{q}}_{\Delta L}]||, \quad (9)$$

where $\mathbf{sg}$ denotes the stop gradient operation, $\beta_1$ and $\beta_2$ are two weight balancing coefficients.

### 3.3 Co-Speech Gesture GPT with Motion Refinement

**Co-Speech Gesture GPT Network.** With the position-irrelevant motion pattern of valid quantization design, each co-speech gesture clip can be transformed into discrete representation. We then learn a co-speech gesture GPT network to map from speech audio $\mathbf{a}_{(1:T)}$ to quantized code sequences $\mathbf{e}^{\mathbf{q}}_{\Delta\mu,(1:T')}$ and $\mathbf{e}^{\mathbf{q}}_{\Delta L,(1:T')}$. Specifically, we extract audio features $\mathbf{a}^{onset}_{(1:T')}$ with onset strength information, which is more suitable for cross-modal pattern learning [46, 49]. Then, a feature embedding layer with positional embedding is leveraged to obtain the tokens for audio onset features, quantized relative shift-translation and qunatized relative factorial covariance. Further, we encode cross-attention information with a series of transformer layers. Finally, followed by a linear transformation with softmax activation, the $M$-dimensional output denotes the probability of each quantization code at that time step. The whole co-speech gesture GPT is trained with cross-entropy loss $\mathcal{L}_{CE}$. Such design enables us to predict and sample future quantization code $\mathbf{e}^{\mathbf{q}}_{\Delta\mu}$ and $\mathbf{e}^{\mathbf{q}}_{\Delta L}$ with speech audio.

**Motion Refinement by Learning Residuals.** Now that we can reconstruct the relative shift-translation $\widehat{\Delta\mu}$ and factorial covariance change $\widehat{\Delta L}$ by VQ-VAE decoding. Given $\mu_1$ and $L_1$ extracted from the initial image frame $I_{(1)}$, the absolute shift-translation and factorial covariance for the $j$-th frame can be calculated as $\widehat{\mu_j} = \mu_1 + \sum_{i=2}^{j}\widehat{\Delta\mu_i}$ and $\widehat{L_j} = L_1 + \sum_{i=2}^{j}\widehat{\Delta L_i}$, respectively. However, since the quantized codebook is designed to only represent the large-scale common motion pattern information, while fine-grained rhythmic details are omitted. Therefore, we propose to refine the co-speech movements by learning residual terms. Concretely, we extract the audio mfcc features $\mathbf{a}^{mfcc}_{(1:T)}$ to encode more contextual audio cues for prosodic dynamics learning. Then a bi-directional LSTM is used to predict the per-frame motion representation residuals $R$ to the main motion pattern result $\widehat{\mu}_{(1:T)}$ and $\widehat{L}_{(1:T)}$, i.e., $\widehat{\mathcal{M}}^{rhythmic}_{(1:T)} = \left[R(\widehat{\mu}_{(1:T)}; \mathbf{a}^{mfcc}_{(1:T)}); R(\widehat{L}_{(1:T)}; \mathbf{a}^{mfcc}_{(1:T)})\right]$. By adding residual terms, the overall co-speech gesture GPT with motion refinement learning can be formulated as:

$$\mathcal{L}_{Residual} = ||\mathcal{M}_{(1:T)} - \widehat{\mathcal{M}}_{(1:T)}(\mathbf{a})||, where \ \widehat{\mathcal{M}}_{(1:T)}(\mathbf{a}) = \widehat{\mathcal{M}}^{pattern}_{(1:T)}(\mathbf{a}^{onset}) + \widehat{\mathcal{M}}^{rhythmic}_{(1:T)}(\mathbf{a}^{mfcc}). \ (10)$$

In this way, we capture both major gesture patterns and subtle rhythmic dynamics for vivid results.

Table 1: **The quantitative results on PATS Image Dataset.** We compare the proposed **A**udio-drive**N** **G**esture v**I**deo g**E**neration (**ANGIE**) against recent SOTA methods [21, 33, 39, 62] and ground truth on four speakers' subsets. For FGD the lower the better, and the higher the better for other metrics.

| Methods | Oliver | | | Seth | | | Kubinec | | | Jon | | |
|---|---|---|---|---|---|---|---|---|---|---|---|---|
| | FGD | BC | Div. | FGD | BC | Div. | FGD | BC | Div. | FGD | BC | Div. |
| GT | 0.00 | 0.76 | 54.6 | 0.00 | 0.71 | 49.3 | 0.00 | 0.84 | 38.9 | 0.00 | 0.73 | 62.8 |
| S2G [21] | 8.57 | 0.59 | 46.1 | 5.75 | 0.62 | 38.2 | 4.76 | 0.67 | 31.6 | 6.07 | 0.51 | 47.3 |
| HA2G [33] | 3.28 | **0.75** | 49.2 | 4.06 | **0.72** | 40.1 | 2.98 | 0.79 | 32.3 | 3.74 | 0.64 | 50.2 |
| SDT [39] | 1.04 | 0.61 | 52.9 | 1.97 | 0.58 | **46.7** | 1.15 | 0.77 | 36.1 | 1.63 | 0.60 | 57.4 |
| TriCon [62] | 3.63 | 0.53 | 48.3 | 3.79 | 0.52 | 40.3 | 3.27 | 0.77 | 35.7 | 3.98 | 0.61 | 49.7 |
| **ANGIE** | **0.88** | 0.72 | **53.5** | **1.83** | 0.69 | **46.7** | **1.10** | **0.81** | **36.5** | **1.57** | **0.65** | **60.9** |

## 4 Experiments

### 4.1 Experimental Settings

**Dataset and Preprocessing.** Pose, Audio, Transcript, Style (PATS) is a large-scale dataset of 25 speakers with aligned pose, audio and transcripts [1, 2, 21]. The training corpus contains 251 hours of data with around 84,000 intervals of mean length 10.7s. Notably, the PATS dataset contains three modalities of audio log-mel spectrograms, speech transcripts and per-frame skeletons labeled with OpenPose [9]. To bypass the error accumulation in pose annotation and facilitate co-speech gesture image generation task, we extend PATS with more features: 1) preprocessed image frames and 2) onset strength audio features which are more suitable for co-speech gesture pattern learning.

We conduct the experiments on four speakers' co-speech video subsets, including Oliver, Seth, Kubinec and Jon. Concretely, 2D skeletons of the image frames are obtained by OpenPose [9] for baseline methods training. The frames are cropped to make the speaker locate at the image center. Since the time span of a meaningful co-speech gesture unit sequence ranges from 4s to 14s [27, 47], we trim invalid videos and finally obtain 1306, 990, 1294 and 1284 clips for four subsets, respectively. The overall mean clip length is 9.8s. We randomly split the segments into 90% for training and 10% for evaluation. The image frames are sampled at 25 fps and further resized to $256 \times 256$.

**Comparison Methods.** We compare with recent SOTA works: 1) *Speech to Gesture* (S2G) [21], a GAN-based pipeline that maps audio to 2D keypoints with a U-Net; 2) *Hierarchical Audio to Gesture* (HA2G) [33] which captures the hierarchical associations between multi-level audio features and tree-like human skeletons; 3) *Speech Drives Template* (SDT) [39] which relieves the one-to-many mapping ambiguity by a set of continuous gesture template vectors; 4) *Trimodal Context* (TriCon) [62], a representative framework that considers the trimodal context of audio, text and speaker identity. Note that all methods could drive 2D human skeletons with speech audio. We train baselines on the PATS image dataset and tune the hyper-parameters by grid search for the best evaluation result. In particular, we also show direct evaluations on the *Ground Truth* (GT) skeletons for clearer comparison.

**Implementation Details.** We sample $T = 96$ frame clips with stride 32 for training. 1) For the VQ-Motion Extractor: the co-speech gesture pattern codebook size $M$ for both relative shift-translation $\Delta\mu$ and factorial covariance change $\Delta L$ are set to 512. The encoders $E_{\Delta\mu}$, $E_{\Delta L}$ and the decoders $D_{\Delta\mu}$, $D_{\Delta\mu}$ are based on 1D-convolution structure. The channel dimension $\ell$ of each codebook entry $\mathbf{d}_{\Delta\mu}$, $\mathbf{d}_{\Delta L}$ as well as the encoded latent features $\mathbf{e}_{\Delta\mu}$, $\mathbf{e}_{\Delta L}$ are 512, while the temporal dimension $T'$ is set as $T/8 = 12$ to encode contextual features with downsampling rate of 8. The $\epsilon$ is set as $1 \times 10^{-5}$ to guarantee the positiveness of diagonal entries in factorial covariance $L$. The commit loss trade-offs in $\mathcal{L}_{VQ}$ are empirically set as $\beta_1 = \beta_2 = 0.1$. We optimize the gesture pattern VQ-VAE with Adam optimizer [28] of learning rate $3 \times 10^{-5}$. 2) For the Co-Speech GPT: the Transformer channel dimension is 768, and the attention layer is implemented in 12 heads with dropout probability of 0.1. The onset strength audio features $\mathbf{a}^{onset} \in \mathbb{R}^{426}$ are extracted by Librosa, while the audio mfcc features $\mathbf{a}^{mfcc} \in \mathbb{R}^{28 \times 12}$ are computed with the window size of 10 ms. During the GPT training, the $\mathbf{e}^{\mathbf{q}}_{\Delta\mu,(1:11)}$, $\mathbf{e}^{\mathbf{q}}_{\Delta L,(1:11)}$ are used as input while $\mathbf{e}^{\mathbf{q}}_{\Delta\mu,(2:12)}$, $\mathbf{e}^{\mathbf{q}}_{\Delta L,(2:12)}$ serve as supervision labels. 3) For the motion representation $\mathcal{M}$ and image generator $G$: we implement as MRAA [45] to use $K = 20$ regions. The motion estimator is pretrained for knowledge distillation. The overall framework is implemented in PyTorch [37] and trained on one 16G Tesla V100 GPU for three days.

Table 2: **User study results on co-speech gesture generation quality.** The rating scale is 1-5, with the larger the better. We compare the *Realness*, *Synchrony* and *Diversity* to baselines [21, 33, 39, 62].

| Methods | GT | S2G [21] | HA2G [33] | SDT [39] | TriCon [62] | **ANGIE (Ours)** |
|---------|------|------|------|------|------|------|
| Realness | 4.29 | 3.27 | 3.92 | 4.01 | 3.74 | **4.08** |
| Synchrony | 4.36 | 3.48 | 4.01 | 3.97 | 3.85 | **4.11** |
| Diversity | 3.97 | 2.49 | 3.31 | 3.88 | 3.02 | **3.95** |

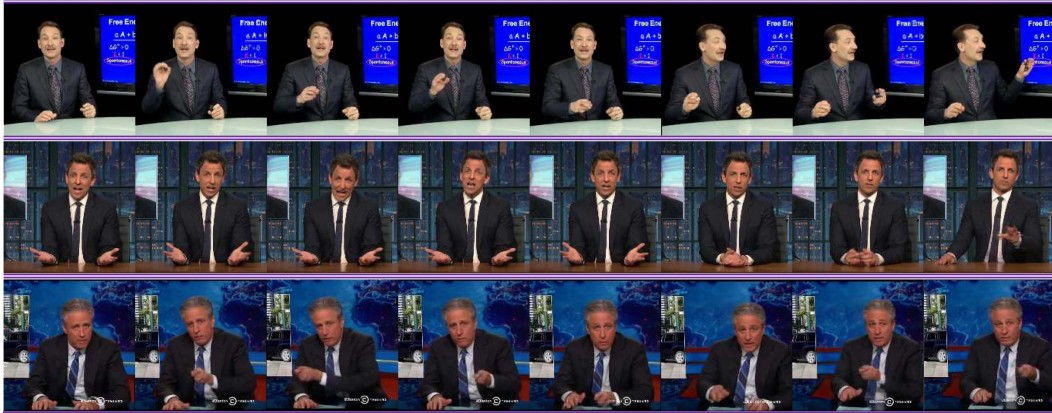

Figure 3: **Image sequence results of ANGIE**. We show the co-speech gesture image generation results of Kubinec, Seth and Jon, respectively. More qualitative results can be found in demo video.

## 4.2 Quantitative Evaluation

**Evaluation Metrics.** We adopt 1) *Fréchet Gesture Distance* (FGD) [62] to evaluate the distance between the real and synthetic gesture distribution. We train an auto-encoder on the PATS image dataset and use the encoder to compute fréchet distance between the real and synthetic gesture in feature space. We also use the 2) *Beat Consistency Score* (BC) and 3) *Diversity* (Div.) [31, 33] to account for the speech-motion alignment and the diversity among generated gestures. Specifically, BC is computed as the average temporal distance between each audio beat and its closest gesture beat, and Diversity indicates the difference of multiple audios' corresponding gestures in the latent space. Note that since all metrics are skeleton-based, we downgrade our method to operate on skeleton data for fair comparison, *i.e.*, we use VQ-VAE w/o cholesky scheme to create 2D skeletons for evaluation.

**Evaluation Results.** The results are reported in Table 1. It can be seen that the proposed ANGIE achieves the best evaluation results on most metrics. Since our method summarizes reusable co-speech gesture patterns into quantized codebooks and complements subtle rhythmic dynamics, we can cover richer gesture patterns and create diverse results. Note that HA2G [33] tends to generate over-expressive gestures with multi-level audio features, which makes their results on BC even better than the ground truth in some cases. Despite of this, we perform comparable to ground truth on BC metric with stable motion results, showing that we can generate audio-aligned gestures. Besides, we can find that both SDT [39] and ours perform better on FGD and Diversity metrics than other methods due to the explicit modeling of co-speech gesture patterns. However, since the gesture pattern is finite and discrete, our quantized codebook design is more suitable than continuous representation.

## 4.3 Qualitative Analysis

**User Study.** We further conduct a user study to reflect the quality of audio-driven gestures. Concretely, we sample 25 audio clips from the PATS image test set for all methods to generate skeleton results, then involve 18 participants for user study. The Mean Opinion Scores protocol is adopted, which requires the participants to rate three aspects: (1) *Realness*; (2) *Synchrony*; (3) *Diversity*. The rating scale is 1 to 5, with 1 being the poorest and 5 being the best. The results are reported in Table 2, where our method performs the best on all three aspects. Notably, with the help of motion pattern codebook,

Table 3: **Ablation study of VQ-Motion Extractor and Co-Speech GPT with Motion Refinement.**

| Ablation Settings | FGD ↓ | BC ↑ | Diversity ↑ | $\mathcal{L}_1$ ↓ | Perceptual ↓ | AED ↓ |
|---|---|---|---|---|---|---|
| w/o Quantization | 5.86 | 0.54 | 35.6 | 0.071 | 79.2 | 0.095 |
| Quantize $\mu$, $L$ | - | - | - | 0.058 | 67.4 | 0.086 |
| Quantize $\Delta\mu$, $L$ | - | - | - | 0.043 | 52.3 | 0.069 |
| Quantize $\mu$, $\Delta L$ | - | - | - | 0.052 | 63.6 | 0.083 |
| w/o Motion Refinement | 1.39 | 0.58 | 48.3 | 0.041 | 48.1 | **0.063** |
| **ANGIE (Ours)** | **1.35** | **0.72** | **49.4** | **0.037** | **42.9** | **0.063** |

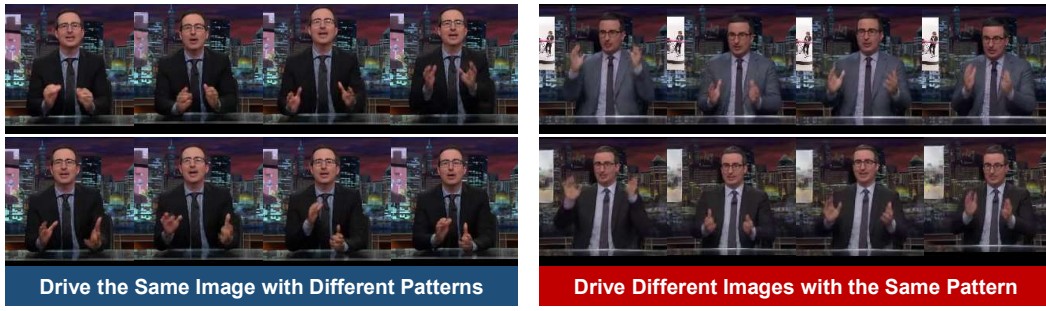

Figure 4: **Codebook Analysis.** We validate that the codebooks contain meaningful motion patterns.

we achieve comparable diversity result to the ground truth, demonstrating that the quantization design helps to capture common gesture patterns.

**Video Generation Results.** With the Co-Speech GPT and motion refinement network, we could predict the future quantization code and complement rhythmic motion details based on audio features, which enables us to generate video results. As shown in Fig. 3, the synthesized image sequence results from left to right contain diverse and meaningful gestures that are aligned with speech audio.

**Codebook Analysis for Co-Speech Gesture Pattern.** We analyze the meaning of the quantized codebooks by asking the question: Does each codebook entry really correspond to a certain type of motion pattern? To answer this, we validate two things in Fig. 4: 1) Different entries represent diverse gesture patterns, hence driving the same image with different quantized codes leads to different gestures (left). 2) Each codebook entry denotes a fixed motion pattern, hence driving different images with the same quantized code show same motions (right). Please refer to demo video for more results.

### 4.4 Ablation Study

In this section, we present ablation study of two modules in our framework. Note that except for the skeleton-based metrics, we also use video reconstruction accuracy as a proxy for image quality, including $\mathcal{L}_1$ and perceptual loss [26] between the reconstructed and GT image; *Average Euclidean Distance* (AED) evaluates identity preservation by pretrained re-identification networks [24, 45].

**VQ-Motion Extractor.** We conduct ablation experiments on five settings: 1) w/o Quantization, where we directly infer motion representation with audio; 2) Quantize $\mu$, $L$; 3) Quantize $\Delta\mu$, $L$; 4) Quantize $\mu$, $\Delta L$, where we quantize the absolute or relative difference of the motion components; 5) w/o Cholesky Decomposition, which means the covariance matrix is not guaranteed to be symmetric positive definite. The results are shown in Table 3, which proves that the quantization design with position-irrelevant relative motion pattern could improve the performance. We also find that quantizing $\Delta\mu$ is more effective than $\Delta L$, since the shift translation is more correlated with region position. Note that when motion component is invalid, the pipeline fails to generate image due to the numerical instability in calculating affine matrix. Hence the results for the setting 5 are not reported.

**Motion Refinement Module.** To verify the efficacy of motion residuals, we eliminate the motion refinement for ablation. The results in Table 3 suggest that this module improves beat consistency by capturing subtle rhythmic movements, while the improvement on FGD majorly derives from the quantization design. Such module complements the motion pattern for fine-grained results.

## 5  Discussion

**Conclusion.** In this paper, we propose a novel framework **ANGIE** to generate audio-driven co-speech gesture video in the image domain. To summarize valid common co-speech gesture patterns, we propose VQ-Motion Extractor with cholesky decomposition based quantization scheme and position-irrelevant design to represent relative motion patterns. Then we propose Co-Speech GPT to refine subtle rhythmic movement details for fine-grained results. Extensive experiments demonstrate that our framework renders realistic and vivid co-speech gesture video generation results.

**Ethical Consideration.** Generating co-speech gesture images facilitates applications such as digital human. However, it could be misused for malicious purposes like forgery generation. We believe that the proper use of this technique will enhance the machine learning research and digital entertainment.

**Limitation and Future Work.** As an early work towards audio-driven co-speech video generation without structural prior, we notice that our method fails for some challenging cases. For example, if the source image is in a large pose, it is hard to generalize well in such an out-of-domain data. We will explore how to develop a model with higher generalization ability in future work.

## 6  Acknowledgment

This work is in part supported by GRF 14205719, TRS T41-603/20-R, Centre for Perceptual and Interactive Intelligence, and CUHK Interdisciplinary AI Research Institute; and in part supported by NTU NAP, MOE AcRF Tier 2 (T2EP20221-0033), and under the RIE2020 Industry Alignment Fund – Industry Collaboration Projects (IAF-ICP) Funding Initiative, as well as cash and in-kind contribution from the industry partner(s).

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
