# Audio-Driven Co-Speech Gesture Video Generation (Supplemental Document)

**Xian Liu**[1], **Qianyi Wu**[2], **Hang Zhou**[1], **Yuanqi Du**[3], **Wayne Wu**[4], **Dahua Lin**[1,4], **Ziwei Liu**[5]

[1]Multimedia Laboratory, The Chinese University of Hong Kong    [2]Monash University
[3]Cornell University    [4]Shanghai AI Laboratory    [5]S-Lab, Nanyang Technological University

In the supplemental document, we will introduce below contents: **1)** proof of Theorem 1 (unique cholesky decomposition theorem) (Sec. A); **2)** more details about Co-Speech Gesture GPT (Sec. B); **3)** model architecture details (Sec. C); **4)** analysis on the limitations and future work (Sec. D); **5)** analysis on the model's generalization ability (Sec. E); **6)** the standard deviation of use study results (Sec. F); **7)** the details of audio feature extraction (Sec. G); **8)** analysis on extra input modality of text information (Sec. H); **9)** experimental results with error bars (Sec. I); **10)** user study details (Sec. J); **11)** ethical considerations with possible prevention measures against potential negative societal impact (Sec. K); **12)** discussions on the dataset consent and personally identifiable information (Sec. L); **13)** the licenses of existing assets involved in this paper (Sec. M);

## A    Proof of Theorem 1

In the main paper, to ease the constraint in the quantization process, we use the *unique cholesky decomposition theorem* [13] to transform the covariance matrix $C$ to factorial covariance $L$ by theorem:

**Theorem 1.** *For any real symmetric positive definite matrix $C \in \mathbb{S}_{++}^n$, there exists a unique lower triangular matrix $L$ with positive diagonal entries, such that $C = LL^\mathsf{T}$.*

**Proof.** As $C$ is a symmetric positive definite matrix, it can be represented as $C = PDP^\mathsf{T}$, where $D$ is a diagonal matrix with positive diagonal entries and $P$ is a unitary matrix. Suppose $B^\mathsf{T} = D^{\frac{1}{2}}P^\mathsf{T}$, it can be factorized as the product of a unitary matrix $Q$ and an upper triangular matrix $R$ (according to Matrix QR Decomposition), *i.e.*, $B^\mathsf{T} = QR$. Thus, we have

$$C = PDP^\mathsf{T} = BB^\mathsf{T} = R^\mathsf{T}Q^\mathsf{T}QR = R^\mathsf{T}R = LL^\mathsf{T}, \tag{1}$$

where $L = R^\mathsf{T}$ is a lower triangular matrix. Because $C$ has the positive determinants, the diagonal entries of $L$ should be non-zero values. Therefore, there exists a lower triangular matrix $L$ with positive diagonal entries such that $C = LL^\mathsf{T}$.

Assume $C = L_1 L_1^\mathsf{T} = L_2 L_2^\mathsf{T}$ where $L_1, L_2$ are two distinct lower triangular matrices with diagonal positive entries, it shows that $I = L_1^{-1} L_2 L_2^\mathsf{T} L_1^{-\mathsf{T}} = (L_1^{-1}L_2)(L_1^{-1}L_2)^\mathsf{T}$ where $I$ is the identity matrix. It means that $L_1^{-1}L_2 = (L_1^{-1}L_2)^{-\mathsf{T}}$ which further shows that $L_1^{-1}L_2$ should be a diagonal matrix. Because $I = (L_1^{-1}L_2)^2$, it demonstrates that $L_1^{-1}L_2$ is a diagonal matrix with $\pm 1$ entries. Since both $L_1, L_2$ have positive diagonal entries, it means $L_1^{-1}L_2 = I$. Therefore, $L_1 = L_2$. $\qquad\square$

## B    More Details about Co-Speech Gesture GPT

The target of the Co-Speech Gesture GPT is to predict the quantized motion codes from the audio onset features $\mathbf{a}^{onset}$. With the proposed vector quantization scheme of position-irrelevant representation, we could transform the raw relative shift-translation $\Delta\mu_{(1:T)}$ and relative factorial covariance change $\Delta L_{(1:T)}$ into corresponding quantized code sequence $\mathbf{e}^{\mathbf{q}}_{\Delta\mu,(1:T')}$ and $\mathbf{e}^{\mathbf{q}}_{\Delta L,(1:T')}$. In this way, any

36th Conference on Neural Information Processing Systems (NeurIPS 2022).

co-speech gesture sequence can be represented by a sequence of quantized motion codes. The original continuous audio-to-gesture prediction task is reframed as to select the proper motion code from the codebooks $\mathcal{D}_{\Delta\mu}$ and $\mathcal{D}_{\Delta L}$ according to the previous motion codes and current speech audio features. The output of the GPT [10] model at the $t$-th time step is the probability of choosing each codebook entry, where the entry with the largest probability serves as the predicted motion code of the next time step.

In particular, given the quantized code sequence $\mathbf{e}^{\mathbf{q}}_{\Delta\mu,(1:T')} \in \mathbb{R}^{T'}$ and $\mathbf{e}^{\mathbf{q}}_{\Delta L,(1:T')} \in \mathbb{R}^{T'}$ of temporal length $T'$, two embedding layers map them to learnable features $\mathbf{f}_{\Delta\mu} \in \mathbb{R}^{T'\times\ell}$ and $\mathbf{f}_{\Delta L} \in \mathbb{R}^{T'\times\ell}$, respectively. The onset audio features $\mathbf{a}^{onset}$ are also passed through an embedding layer to obtain audio features $\mathbf{f}_a \in \mathbb{R}^{T'\times\ell}$. After the concatenation over the temporal dimension and supplement the position embedding information, we get the latent input features of shape $(3 \times T') \times \ell$. Then, 12 Transformer layers with channel dimension of 768 and 12 attention heads of dropout probability 0.1 are utilized to encode the cross-attention information. Finally, a linear and softmax layer map the encoded latent features to normalized motion probabilities of $\mathbf{p} \in \mathbb{R}^{(3\times T')\times M}$, where $M$ is the codebook size. Notably, $\mathbf{p}_{t,m}$ denote the probability of motion code $\mathbf{d}_{\Delta\mu,m}$ and $\mathbf{d}_{\Delta L,m}$ predicted at the time step $t+1$ for relative shift-translation $\Delta\mu$ and relative factorial covariance change $\Delta L$, respectively. During the implementation, the probabilities at the audio feature indexes $\mathbf{p}_{(1:T')}$ are discarded, while the probabilities at the $\Delta\mu$ and $\Delta L$ indexes represent the corresponding motion probabilities, *i.e.*, $\mathbf{p}_{\Delta\mu} = \mathbf{p}_{(T'+1:2T')}$ and $\mathbf{p}_{\Delta L} = \mathbf{p}_{(2T'+1:3T')}$ [12]. The attention layer can be formulated as:

$$\text{Attention}(\mathbf{Q}, \mathbf{K}, \mathbf{V}, \mathbf{M}) = \text{softmax}(\frac{\mathbf{Q}\mathbf{K}^{\mathsf{T}} + \mathbf{M}}{\sqrt{\ell}})\mathbf{V}, \tag{2}$$

where $\mathbf{Q}, \mathbf{K}, \mathbf{V}$ are the query, key and value matrix from input, $\mathbf{M}$ denote the mask, $\ell$ is the channel dimension and $\mathsf{T}$ is the matrix transpose operation. To encode the cross-conditional features of audio features, relative shift-translation features and relative covariance change features, the mask $\mathbf{M}$ is set as a $3 \times 3$ repeated block matrix with the lower triangular matrix as its element.

The Co-Speech GPT is trained via cross-entropy loss as a classification problem. Specifically, the motion codes of $\mathbf{e}^{\mathbf{q}}_{\Delta\mu,(1:11)}$, $\mathbf{e}^{\mathbf{q}}_{\Delta L,(1:11)}$ are used as input, while $\mathbf{e}^{\mathbf{q}}_{\Delta\mu,(2:12)}$, $\mathbf{e}^{\mathbf{q}}_{\Delta L,(2:12)}$ serve as supervision labels. The overall loss function is:

$$\mathcal{L}_{CE} = \frac{1}{T'} \sum_{t=1}^{T'} \left\{ \texttt{CrossEntropy}\left[\mathbf{p}_{(\Delta\mu,t)}, \mathbf{e}^{\mathbf{q}}_{(\Delta\mu,t+1)}\right] + \texttt{CrossEntropy}\left[\mathbf{p}_{(\Delta L,t)}, \mathbf{e}^{\mathbf{q}}_{(\Delta L,t+1)}\right] \right\}, \tag{3}$$

where $\texttt{CrossEntropy}$ denotes the cross-entropy loss function; $\mathbf{p}_{(\Delta\mu,t)}$, $\mathbf{p}_{(\Delta L,t)}$ are the predicted probability of $\Delta\mu$, $\Delta L$ terms at the $t$-th time step, respectively; the $\mathbf{e}^{\mathbf{q}}_{(\Delta\mu,t+1)}$, $\mathbf{e}^{\mathbf{q}}_{(\Delta L,t+1)}$ are the vector quantized code of $\Delta\mu$ and $\Delta L$ at the $(t+1)$-th time step, respectively.

## C  Architecture Details

**VQ-Motion Extractor Encoder and Decoder.** The encoders $E_{\Delta\mu}$, $E_{\Delta L}$ and the decoders $D_{\Delta\mu}$, $D_{\Delta\mu}$ are 1D temporal convolution networks. In particular, the encoders have three convolution layers, with each layer's kernel size of 3, padding of 1 and stride of 2. In this way, the downsampling scale of the encoder is 8, which will compress the temporal contextual information. For the decoder, the structure is similar, with three convolution layers of kernel size 3, padding 1 and stride 1, and three upsample layers of upsample scale 2 to reconstruct the latent features.

**Motion Estimator and Image Generator.** For the motion estimator, we follow MRAA [11] that uses a region predictor to extract the articulated human body information. Specifically, we similarly use the U-Net architecture with five "`convolution - batch norm - ReLU - pooling`" blocks in the encoder and five "`upsample - convolution - batch norm - ReLU`" blocks in the decoder for both the region predictor and pixel-wise flow predictor. Note that since it is inappropriate to infer the background motion from speech audio cues, we remove the original background motion predictor. In terms of the image generator, it is based on the Johnson architecture [5], with two down-sampling blocks, six residual-blocks, and two up-sampling blocks. The skip connections are added to warp and weight the confidence maps.

Table 1: **Detailed structure of audio encoder.** [†]Note that in the table, Conv2d ($c_{in}$, $c_{out}$, $k$, $s$, $p$) means the conv2d operation of input channel dimension $c_{in}$, the output channel dimension of $c_{out}$, the kernel size of $k$, the stride of $s$ and the padding of $p$; MaxPool2d ($k$, ($s_h$, $s_w$)) means the MaxPool2d operation of kernel size $k$, h direction stride of $s_h$ and w direction stride of $s_w$; Linear ($c_{in}$, $c_{out}$) means the linear layer with input feature of dimension $c_{in}$ and output feature of dimension $c_{out}$; ReLU means the ReLU operation on the input feature.

| Audio Encoder | | |
|---|---|---|
| Feature | Feature Shapes | Operations |
| Input | $1 \times 28 \times 12$ | Conv2d (1, 64, 3, 1, 1) |
| Layer-1 | $64 \times 28 \times 12$ | Conv2d (64, 128, 3, 1, 1) |
| Layer-2 | $128 \times 28 \times 12$ | MaxPool2d (3, (1, 2)) |
| Layer-3 | $128 \times 26 \times 5$ | Conv2d (128, 256, 3, 1, 1) |
| Layer-4 | $256 \times 26 \times 5$ | Conv2d (256, 256, 3, 1, 1) |
| Layer-5 | $256 \times 26 \times 5$ | Conv2d (256, 512, 3, 1, 1) |
| Layer-6 | $512 \times 26 \times 5$ | MaxPool2d (3, (2, 2)) |
| Flatten | $512 \times 12 \times 2$ | flatten |
| Layer-7 | 12288 | Linear (12288, 2048), ReLU |
| Layer-8 | 2048 | Linear (2048, 256), ReLU |
| Layer-9 | 256 | Linear (256, 128), ReLU |
| Per-frame Audio Feature | 128 | - |

**Motion Refinement Network.** The motion refinement network maps the audio mfcc features to the residual terms to complement the subtle rhythmic movements of the co-speech gesture. Concretely, the audio mfcc features $\mathbf{a}^{mfcc} \in \mathbb{R}^{28 \times 12}$ are first pre-extracted with the window size of 10 ms. Then a convolution based audio encoder with a series of linear layers is used to encode the per-frame audio features into $\mathbf{f}_a^{mfcc} \in \mathbb{R}^{128}$. The detailed structure is listed in Table 1.

We first follow the pipeline of MRAA [11] to pretrain the motion estimator and image generator via self-reconstruction. Then, the motion estimator module is freezed and serves as supervision in VQ-Motion Estimator training, *i.e.*, we train the VQ-VAE model to reconstruct the motion representation from pretrained MRAA motion estimator.

# D    Analysis on Limitations and Future Work

As an early work towards audio-driven co-speech video generation without structural prior, we notice that our method fails for some challenging cases. For example, if the source image is in a large pose, it is hard to generalize well in such out-of-domain data.

Notably, since our work does not involve any structural human body prior, it is of high freedom without any human skeleton physical constraint. Therefore, for some cases of large pose, the model may generate too large motion field and lead to failure case results. We will explore how to develop a model with higher generalization ability in future work.

Generalizing co-speech gesture avatars to more complex and general settings like social conversation is a promising idea of great practical usage. Currently, the biggest bottleneck could be the lack of high-quality conversational dataset. Although CMU Panoptic contains the multi-view conversation videos, the image quality is poor for co-speech image animation. Besides, since the social co-speech gesture is more diverse, some model designs like VQ codebook size should be well studied. We will delve into this interesting problem in future work.

Besides, since our frames are preprocessed to resolution of $256 \times 256$, the scale and resolution of facial regions are rather small. We leverage Wav2Lip [8] to sync the lip shape in the demo video for better visualization. Note that such process substantially differs from the post-processing step of [4, 9] that trains an extra pose2image generator: **1)** The focus of co-speech gesture generation task is to synthesize coherent upper body poses that are aligned to speech audio, while the lip-sync is not the focus in this work. We merely implement it for better visualization. **2)** We could directly use the off-the-shelf Wav2Lip [8] to sync the lip shape without any preprocessing step, post-processing step or re-training. Therefore, it is in essence different from an additional pose2image generator

Table 2: **Experiment results when generalize to unseen audio.**

| Methods | FGD ↓ | BC ↑ | Diversity ↑ |
|---|---|---|---|
| ANGIE (Novel Audio) | 1.46 | 0.69 | 48.5 |
| **ANGIE (Ours)** | **1.35** | **0.72** | **49.4** |

Table 3: **The standard deviation of the user study results.** The rating scale is 1-5, with the larger the better. We compare the *Realness*, *Synchrony* and *Diversity* to baselines [4, 6, 9, 15].

| Methods | GT | S2G [4] | HA2G [6] | SDT [9] | TriCon [15] | **ANGIE (Ours)** |
|---|---|---|---|---|---|---|
| Realness | 0.492 | 0.413 | 0.293 | 0.206 | 0.385 | 0.312 |
| Synchrony | 0.480 | 0.574 | 0.214 | 0.265 | 0.350 | 0.345 |
| Diversity | 0.252 | 0.629 | 0.471 | 0.313 | 0.221 | 0.287 |

that requires tedious re-training. **3)** The task of simultaneously generating audio-consistent upper body gestures and lip shapes is challenging and remains unsolved. The difficulty lies in **a)** the low-resolution of mouth regions in the large-scale upper body image makes it hard to capture the fine-grained lip-sync information; **b)** the mouth shape only depends on the phoneme, while the co-speech gesture correlates to the semantic meanings. We will explore this problem in future work.

# E    Analysis on Generalization Ability

As shown in previous studies [2, 4], the co-speech gesture motions and styles vary a lot for different speakers, which is termed as "individual speaking style" [4]. Therefore, it is suitable to train a separate co-speech gesture generation model for each person following the experiment settings of baselines [2, 4, 9]. However, we explore a more challenging task of co-speech gesture video generation in a unified framework without structural prior and achieve superior performance. Even for a single-person subset, it is non-trivial to animate non-rigid human body in image space by speech audio, especially with the interference of complex background scenes.

It is hard to generalize to speakers that are not in the training data with **currently available co-speech gesture image datasets**. In particular, the commonly used datasets are TED Gesture [15] and PATS [2, 4]. TED Gesture is based on TED Talk videos, while PATS contains 25 speakers of talk shows, lectures, etc. Due to the frequent camera movements and viewpoint shift in TED videos, there lacks clear co-speech gesture clips for **image** generation. Hence we narrow down the experiments to PATS dataset in this work. A dataset with high-quality co-speech gesture image frames of multiple speakers is needed to learn a model of novel (unseen) person generalization ability. We will strive for this in future work.

We verify the potential generalization ability of our approach: **a)** We could animate different appearances of a speaker with speech audio (as shown in codebook analysis part of the demo video, we could animate Oliver's different appearances), while previous studies [9, 4] that resort to off-the-shelf pose2img generator only support a single appearance. **b)** We additionally implement the experiments of animating with novel (unseen) audio. The evaluated results are reported in Table 2. It shows that the model's performance is still effective with the input of unseen audio. With the proposed vector quantize design, each codebook entry corresponds to a reasonable co-speech gesture pattern. In contrast to directly mapping to the continuous coordinate space, such technical design guarantees the valid gesture even when generalizing to the unseen audio.

# F    The Standard Deviation of User Study

We additionally provide the standard deviation of user study results in Table 3, which shows that the agreement among participants is highly consistent.

Table 4: **Experiment results of concatenating text features.**

| Methods | FGD ↓ | BC ↑ | Diversity ↑ |
|---|---|---|---|
| ANGIE (Extra Text Concat) | **1.30** | **0.73** | **49.8** |
| **ANGIE (Ours)** | 1.35 | 0.72 | 49.4 |

## G  Details on Audio Feature Extraction

**Audio Onset Strength Feature.** The audio onset strength feature of $T$ frames is of shape $(T, 426)$, where $T$ is the temporal dimension (frame number) and $426$ is the feature channel dimension. It is the concatenation of constant-Q chromagram, tempogram, onset beat, onset tempo and onset strength. Most features are derived from the audio onset strength/envelope and the channel dimension is summed up to $426$. We utilize the librosa onset functions to extract the features. The audio sample rate is $16000$, the time lag for computing differences is $1$, the hop length is $512$ and the window length is $384$.

**Audio MFCC Feature.** The original audio mfcc was calculated as 12 mfccs which has the dimension of $(T', 12)$ where $T'$ is the original audio frame number. In our implementation, we use a 28-dim sliding window to further unfold the mfcc feature into a final shape of $(T, 28, 12)$, where $T$ denotes the final temporal dimension (video frame number), 28 denotes the size of sliding window and 12 is the mfcc feature dimension. MFCC feature is extracted with a sample rate of 16000, window length of 25 ms, window step of 10 ms, cepstrum number of 13, filters number of 26 and FFT block size of 512. In the motion refinement module, we use the a certain frame's mfcc feature of shape $(28, 12)$ and forward a series of convolution and linear layers to extract the per-frame audio feature of dimension 128.

## H  Analysis on Extra Input Modality of Text Information

Our main focus and technical contributions are how to animate the co-speech gesture in the image domain, but not on the influence of input modality. We choose the audio-driven setting mainly for two reasons: **a)** We generally follow the problem setting of baselines [2, 4, 9], where all the compared methods take only the speech audio as input modality for fair comparison. **b)** We have to pre-process the raw transcripts to be temporally aligned with audio using tools like Gentle. To prevent the potential alignment inaccuracy in the pre-processing step, and to simplify the problem setting for better focusing on audio-driven co-speech gesture image animation, we do not involve the text input in this work.

As proved in Automatic Speech Recognition (ASR) [16, 14] and recent co-speech gesture studies [6], the speech audio actually contains some high-level semantic information. Such implicit semantic information in the speech audio could guide the model to capture some specific co-speech gesture patterns like metaphorics [6]. Besides, as shown in the Kubinec chemistry lecture setting, our model manages to synthesize deictic gestures of pointing to the screen by learning such implicit audio-gesture correlations. We would like to respectfully claim that the generated gestures of our model are not only limited to the beat gestures, but some semantic gestures could also be synthesized.

Typically, language model/text information contains rich semantics, which are beneficial to the learning of semantics relevant gestures like iconics, metaphorics, and deictics. Previous study [15] has also verified the influence of each input modality on co-speech gesture, including audio, text and speaker identity, etc. Therefore, we additionally complement an experiment of using text feature: we encode the Gentle aligned transcripts by TextTCN [3] and further concatenate with audio features. The combined audio text features are fed to Co-Speech GPT with Motion Refinement network to predict the quantized code as well as motion residuals. The results are reported in Table 4, which suggest that the text feature could indeed facilitate the co-speech gesture generation.

As an early attempt exploring audio-driven co-speech video generation, this work could serve as a baseline for further studies in the research community. However, how to effectively fuse the multiple modality information (including audio and text) and better map to the implicit motion representation remains an open problem. We will explore this in future work.

Table 5: **Randomness of the Diversity metric on the Oliver subset.**

| Group | 1 | 2 | 3 | 4 | 5 |
|---|---|---|---|---|---|
| Diversity | 53.29 | 52.88 | 53.62 | 53.59 | 53.36 |

Table 6: **Licenses of existing assets we have used in this work.**

| Asset | License Link |
|---|---|
| MRAA [11] | https://github.com/snap-research/articulated-animation/blob/main/LICENSE.md |
| Librosa [7] | https://github.com/librosa/librosa/blob/main/LICENSE.md |
| Bailando [12] | https://github.com/lisiyao21/Bailando/blob/main/LICENSE |
| PATS [1, 2, 4] | https://github.com/chahuja/pats |

## I  Experimental Results with Error Bars

Since the Diversity metric involves randomness, which may fluctuate due to different sampling clips. Therefore, we randomly sample 400 pairs to get robust evaluation results. To verify the steadiness, we conduct the evaluation 5 times (create random samples 5 times with different random seeds). The results are listed in Table 5. We can see that the fluctuation is rather small, which shows the robustness of such evaluation.

## J  User Study Details

The study involves 18 participants of 9 females and 9 males. In particular, the users are unaware of which generated result corresponds to which method for a fair comparison. Before they rate the quality of synthesized results, the participants will first be shown what the Ground Truth (original raw video) is to help them make accurate scoring. The participants are asked to judge the three perspectives of the generated portraits: (1) *Realness*; (2) *Synchrony*; (3) *Diversity*. All participants are paid for 15 USD for about $40 - 60$ minutes' rating process. We use the Fleiss's-Kappa statistic to measure the participants' scoring disagreement, which is a statistical measure for assessing the reliability of agreement between a fixed number of raters when assigning categorical ratings to a number of items or classifying items.

## K  Ethical Considerations

Our method could synthesize co-speech gesture images, which is envisioned to facilitate extensive applications like digital human and human-machine interaction. On the other hand, however, such techniques could be misused for malicious purposes such as forgery generation. As part of our responsibility, we strongly advocate all safeguarding measures against malicious use of co-speech gesture images and feel obliged to share our generated results with the deepfake detection community to improve the method's robustness. We believe that the proper use of this technique will enhance positive societal development in both machine learning research and human's daily entertainment.

## L  Discussions on Dataset Consent and Information

The PATS Image Dataset is collected and preprocessed based on the PATS dataset. We download the YouTube videos by the provided video links, shared under the "CC BY - NC - ND4.0 International" license. This license allows for non-commercial use. Since all our code, data, models and results are kept for academic use only, hence despite of the contained personally identifiable information in the collected data, we will use them for research use only and will not allow any commercial use.

## M  Assets License

In the Table 6, we list the licenses of all the existing assets we have used in this work.