# OpenReview forum: "Audio-Driven Co-Speech Gesture Video Generation"
_NeurIPS.cc/2022/Conference — NeurIPS 2022 Accept_

### Official Review · Reviewer_WZ3S · 2022-07-10

**Rating:** 7
**Confidence:** 3
**Soundness:** 3 good
**Presentation:** 3 good
**Contribution:** 2 fair

**Summary:**

This paper works on audio-driven gesture image generation, which receives a still image and audio to generate a sequence of images (video). Compared to previous works, it does not rely on skeletal pose annotation to prevent error accumulation caused by cascading of synthesis networks. The proposed system (named as ANGIE) is consisted of VQ-motion extractor and Co-speech GPT, each mapping the image to vector quantized representation and transformer network autoregressively predict next frame's quantized representation, respectively.

**Questions:**

- Regarding the line 272 - "The motion estimator is pretrained for knowledge distillation" - please elaborate on this. The authors might want to describe such implementation details on the appendix.
- What is a new dataset that the authors have collected, as described in line 92? Is it simply an extension of PATS dataset by some post-processing steps?

**Limitations:**

It would be better if the authors could provide specific examples of the limitation of ANGIE (discussed in line 344).

**Strengths And Weaknesses:**

Strenghts
- By vector-quantizing the images, the proposed network could enjoy powerful capability of the Transformer. This kind of framework might be also applied at various video generation networks.
- The motion refinement network to aid missing details of reconstructed images from quantized representation seems novel.
- Ablation studies (Table 3) clearly justifies the choice of the components.

Weaknesses
- The related works shown in section 2 is limited in the application viewpoint. It would be much better if authors could make a connection to slightly other domains that the proposed method could be also applied. In overall, the scope of the paper is too specific on the co-speech gesture image generation. To meet broader NeurIPS readership, I wish there would be some implications outside the main problem.
- Please do not cite Wikipedia article as a reference (line 303, Fleiss's Kappa statistic). Please briefly describe about the concept in the appendix.

---

> ### Author Response · Authors · 2022-08-02
> **Response to Reviewer WZ3S**
>
> We sincerely thank the reviewer for your insightful comments and recognitions to this work, especially for acknowledging that our approach is novel with potential benefits to the community and relevant research domains. We have polished the paper and made the clarifications in the revised version.
>
> The technical contributions and novelty of this work are highlighted in the General Response. Please kindly refer to it for details. Note that the following polishments have been made according to your advice:
>
> * We have polished to highlight the potential impact of our work on relevant research domains in Section “Related Work” of the main submission.
> * We have eliminated the Wikipedia article reference (in L301 of the main submission) and elaborated the concept of Fleiss's Kappa statistic in the supplemental document (in Section J of the supplemental document).
> * We have included the motion estimator training details in Section C of the supplemental document.
>
> Thanks again for your very constructive comments, which have helped us improve the quality of the paper significantly! Below we would like to provide point-to-point responses to all the raised questions:
>
> > **Q1: "The related works shown in section 2 is limited in the application viewpoint. It would be much better if authors could make a connection to slightly other domains that the proposed method could be also applied. In overall, the scope of the paper is too specific on the co-speech gesture image generation. To meet broader NeurIPS readership, I wish there would be some implications outside the main problem."**
>
> **A1:** Thank you for the precious advice! We have polished the writing of related work part in the main submission accordingly. Though focusing on the specific task of co-speech gesture generation in this paper, the novel constrained vector quantization design and residual refinement idea could potentially benefit relevant research domains like constrained VQ problem and video generation. We sincerely thank the reviewer for appreciating this!
>
> > **Q2: "Please do not cite Wikipedia article as a reference (line 303, Fleiss's Kappa statistic). Please briefly describe about the concept in the appendix."**
>
> **A2:** Many thanks for your suggestion! We have polished this part in the revised version. We eliminate the Wikipedia article reference (L301) and elaborate the concept of Fleiss's Kappa statistic in the supplemental document (Section J).
>
> > **Q3: "Regarding the line 272 - "The motion estimator is pretrained for knowledge distillation" - please elaborate on this. The authors might want to describe such implementation details on the appendix."**
>
> **A3:** We first follow the pipeline of MRAA [a] to pretrain the motion estimator and image generator via self-reconstruction. Then, the motion estimator module is freezed and serves as supervision in VQ-Motion Estimator training, i.e., we train the VQ-VAE model to reconstruct the motion representation from the pretrained MRAA motion estimator. In this way, the model gradually learns the knowledge of the pretrained MRAA motion estimator, which resembles the process of knowledge distillation. We have included the implementation details in Section C of the supplemental document.
>
> > **Q4: "What is a new dataset that the authors have collected, as described in line 92? Is it simply an extension of PATS dataset by some post-processing steps?."**
>
> **A4:** We complement the original PATS dataset with more processed features, including: **1)** pre-processed image frames and **2)** onset strength audio features (as detailed in L238 of the main submission). These new features are important to the co-speech gesture image generation task. We hope this could facilitate future research in the community.
>
> > **Q5: "It would be better if the authors could provide specific limitation examples (discussed in line 344)."**
>
> **A5:** Many thanks for your advice! Due to the difficulty of this challenging task, ANGIE fails to generate human images of extreme pose. Besides, with the lack of high-resolution co-speech upper body images, in some cases ANGIE fails to synthesize the high-fidelity face and upper body simultaneously. Please kindly refer to Section D of the supplemental document for specific examples of the limitation, future work and discussions.
>
> ****
>
> [a] - Siarohin et al. "Motion Representations for Articulated Animation."
>
> ****
>
> Please don’t hesitate to let us know if there are any additional clarifications or experiments that we can offer!

---

> ### Author Response · Authors · 2022-08-05
> **Additional Response to Reviewer WZ3S**
>
> Dear Reviewer WZ3S:
>
> We sincerely thank you again for your great efforts in reviewing this paper, especially for the precious advice that has helped us improve the quality of this paper significantly!
>
> We have polished the related work, updated the citation and included the limitation examples in the revised version. Please don't hesitate to let us know if there are further clarifications or experiments that we could offer!
>
> Best,
>
> Paper 365 Authors.

---

> > ### Comment · Reviewer_WZ3S · 2022-08-09
> > **Thank you for your response**
> >
> > Dear Paper365 Authors:
> >
> > Thank you for the response. All of my concerns have been addressed, so I'll be updating my rating accordingly.

---

> > > ### Author Response · Authors · 2022-08-09
> > > **Thanks for your comments!**
> > >
> > > Dear Reviewer WZ3S:
> > >
> > > We are delighted to hear that your concerns are addressed! Many thanks again for your very constructive comments, which have helped us improve the quality of the paper significantly.
> > >
> > > Best,
> > >
> > > Paper 365 Authors.

---

### Official Review · Reviewer_S12n · 2022-07-11

**Rating:** 6
**Confidence:** 4
**Soundness:** 3 good
**Presentation:** 3 good
**Contribution:** 3 good

**Summary:**

This paper concerns synthesis of co-speech gestures, gestures are coded in terms of entries in a VQ code book, and a GPT-like model is used to generate the sequence of symbols corresponding to input audio. In the generation, first the broad gestures are created, and then these are refined to improve the overall fidelity.

**Questions:**

The audio clips are variable length, are these clipped to be a fixed size?

How do you ensure consistency when generating longer sequences? Only short segments are shown in the demo video, it would be nice to see longer generated sequences.

I was confused by “position-irrelevant” because position is important as position gives the sense of scale in gesture space. It was cleared up in later text that you mean image location invariant.

In the demo video, the speaker refers to “four quadrants” and it looks like the model produces gestures that seem to point to four quadrants. This seems suspicious since the “common motion patterns” are generated from just the speech envelope, and the refined motion from the MFCCs.


**Limitations:**

Clarify that the approach is specifically targeting only beat gestures.

**Strengths And Weaknesses:**

+ To construct the training data, an existing dataset was augmented with new features, and this will be made available.
+ The generated image sequences look compelling.
+ The approach was compared against, and beat, several baselines.

- The paper refers to gestures as “common motion patterns” and “rhythmic dynamics”.  What do these mean?  Use standard terminology from the co-speech gesture literature.
- It looks like the models are over fit to the specific sequences.  For example, I think it’s the Kubinec sequence: when the speaker is referring to information on the screen to the speaker’s left, the speaker points and gestures towards the screen.  Why?  Is the screen always to the speaker’s left?  There is no language model and there is no explicit capturing of the speech content, so it is not clear to me why we would see this behavior.
- I believe you are training a separate model for each speaker, but you criticized prior work for limited generalization ability.
- The paper considers gesture only in a very limited context - talk show host speaking monologue to camera.  How well do you expect this to generalize, e.g., to conversational gestures that are likely more subtle and drawn from a larger lexicon?
- There are broad categories of gesture that this approach will fail to capture.  Specifically without the use of language, iconics, metaphorics, and deictics, are unlikely to be generated.  You ought to clarify that the focus is specifically only on beats.

---

> ### Author Response · Authors · 2022-08-02
> **Response to Reviewer S12n [Part 3/3]**
>
> > **Q1: "The paper refers to gestures as “common motion patterns” and “rhythmic dynamics”. What do these mean? Use standard terminology from the co-speech gesture literature."**
>
> **A1: 1)** The “common motion patterns” determine the general appearance of gestures in a generated sequence, while the “rhythmic dynamics” mean the subtle movements to match the speech audio [a, b]. Previous studies refer to the common motion pattern as similar terms like “motion template” [a] or “pose mode” [b].
>
> **2)** In this work, with the vector quantize design, we could extract the reusable motion patterns from training data as codebook entries. Since those gestures of similar general appearance tend to be mapped into the same quantized code, we refer to such motion patterns as “common”.
>
> > **Q2: "Seems overfit to specific sequences. Why does the speaker points towards the screen in Kubinec? Is the screen always to the speaker’s left? No language model/speech content, why we would see this behavior."**
>
> **A2:** Please kindly refer to the Common A2.
>
> > **Q3: "Train a separate model for each speaker, but criticized prior work for limited generalization ability."**
>
> **A3:** Please kindly refer to the Common A3.
>
> > **Q4: "Consider gesture in a very limited context - talk show host speaking monologue to camera. How well could generalize to conversational gestures that are likely more subtle and drawn from a larger lexicon?"**
>
> **A4:** Thank you for the precious advice. Generalizing co-speech gesture avatars to more complex and general settings like conversation is a promising idea of great practical usage. Currently, the biggest bottleneck could be the lack of high-quality conversational image dataset. Although CMU Panoptic [j] contains multi-view conversation videos, the image quality is poor for co-speech image animation. Besides, since the social co-speech gesture is more diverse, some model designs like VQ codebook size should be well studied. We will delve into this interesting problem in future work. The discussions are included in Section D of the supplemental document.
>
> > **Q5: "Fail to cappture iconics, metaphorics, and deictics without text. Only beat gesture."**
>
> **A5:** Please kindly refer to the Common A1.
>
> > **Q6: "The audio clips are variable length, are these clipped to be a fixed size?"**
>
> **A6:** At the audio pre-processing step, they are not clipped to a fixed size. At the training stage, since we have to input a certain length of audio to the model, we sample a sliding window of 96 frame audio clip with stride of 32. The details are elaborated in L256 of the main submission.
>
> > **Q7: "Consistency when generating longer sequences."**
>
> **A7: 1)** The long-term consistency is guaranteed by Co-Speech GPT, where the attention mechanism in transformer model would take the self-attention of previous gestures and cross-attention of input speech audio for coherent results. We will include longer sequence results in the final version.
>
> **2)** Compared to previous setting of generating co-speech gesture skeleton, animating long sequence in the image domain is harder. Actually, generating long sequence remains an open problem in video generation, let alone the more challenging cross-modal audio-to-gesture generation. We will keep exploring how to generate the long-sequence results in future work.
>
> > **Q8: "Position-irrelevant. It was cleared up in later text that you mean image location invariant."**
>
> **A8:** Yes, your understanding is correct! We have polished accordingly in the revised version.
>
> > **Q9: "Only use the speech envelope and MFCCs to generate gestures that seem to point to four quadrants."**
>
> **A9:** Please kindly refer to the Common A2.
>
> > **Q10: "Clarify that the approach is only for beat gestures."**
>
> **A10:** Please kindly refer to the Common A1.
>
> ****
>
> [a] - Qian et al. "Speech Drives Templates: Co-Speech Gesture Synthesis with Learned Templates."
>
> [b] - Xu et al. "Freeform Body Motion Generation from Speech."
>
> [c] - Ginosar et al. "Learning Individual Styles of Conversational Gesture."
>
> [d] - Ahuja et al. "Style Transfer for Co-Speech Gesture Animation: A Multi-Speaker Conditional-Mixture Approach."
>
> [e] - Yu et al. "Audio-Visual Recognition of Overlapped Speech for the Irs2 Dataset."
>
> [f] - Winata et al. "Lightweight and Efficient End-to-end Speech Recognition Using Low-rank Transformer."
>
> [g] - Liu et al. "Learning Hierarchical Cross-Modal Association for Co-Speech Gesture Generation."
>
> [h] - Yoon et al. "Speech Gesture Generation from the Trimodal Context of Text, Audio, and Speaker Identity."
>
> [i] - Bai et al. "An Empirical Evaluation of Generic Convolutional and Recurrent Networks
> for Sequence Modeling."
>
> [j] - Joo et al., "Panoptic Studio: A Massively Multiview System for Social Interaction Capture."
>
> ****
>
> Please don’t hesitate to let us know if there are any additional clarifications or experiments that we can offer!

---

> > ### Comment · Reviewer_S12n · 2022-08-03
> > **Thank you for the detailed response**
> >
> > Thank you for the detailed responses to my concerns.  I appreciate the time you have taken to address the issues and run the additional experiments.  In light of this, the opinions of the other reviewers, and your response to their reviews, I am satisfied that you have addressed my concerns. I will revise my decision accordingly.

---

> > > ### Author Response · Authors · 2022-08-03
> > > **Thanks for your comments!**
> > >
> > > Dear Reviewer S12n:
> > >
> > > We are delighted to hear that your concerns are addressed! Many thanks again for your very constructive comments, which have helped us improve the quality of the paper significantly.
> > >
> > > Best,
> > >
> > > Paper 365 Authors.

---

> > > ### Author Response · Authors · 2022-08-05
> > > **A Kind Reminder to Reviewer S12n**
> > >
> > > Dear Reviewer S12n:
> > >
> > > Sorry for the bothering. We are very delighted that your concerns have been addressed and we sincerely thank your comments for acknowledging that our responses are satisfactory! We would like to kindly remind that your rating of this paper seems unchanged. If you have any further question, please let us know. Many thanks again!
> > >
> > > Best,
> > >
> > > Paper 365 Authors.

---

> ### Author Response · Authors · 2022-08-02
> **Response to Reviewer S12n [Part 2/3]**
>
> > **Common Q2: "Since no text or language models is used, why could we generate “pointing to screen” gesture in Kubinec subset? Is the screen always to the speaker’s left? Seems overfit to the specific sequences."**
>
> **Common A2: 1)** In the Kubinec chemistry lecture setting, the screen is always to the speaker’s left. Such dataset bias indeed exists, which is verified and visualized by previous work [c] (please kindly refer to the Figure 2 of [c] for the gesture heatmap). They refer to such phenomenon as “individual speaking style”, where the behavior of frequently pointing to the screen is the speaker style learned by the model, but not overfitting [a, c, d].
>
> **2)** Since the reference video of the same audio does not show such behavior, our model is neither overfitting nor memorizing the specific hand location (otherwise, the ground truth will point to the same location as well). We further analyze which module leads to such phenomenon by visualizing the generated results of each module. We find that the speaker’s hand is pointing to the screen (not exactly on the “four quadrants”) after the vector quantize module, while the motion refinement module refines the height of hand, so that the hand is pointing to that location. This shows that the VQ network determines a certain motion pattern of pointing to screen, and residual learning refines the hand moving height, which demonstrates the effectiveness of our method.
>
> **3)** As mentioned in the **Common A1 (2)**, the speech audio indeed contains some high-level semantic information. With the proposed vector quantization design, it simplifies the learning from a harder regression problem to an easier classification problem (as detailed in L172-174 of the main submission). In this way, we ease the difficulty of learning such a cross-modal mapping. The model could thus better grasp the connections between speech audio and the “point to the screen” gesture.
>
> > **Common Q3: "Separate model for each speaker, but criticize prior work for limited generalization ability."**
>
> **Common A3: 1)** We would like to clarify that our meaning of “limited generalization ability” is under the context of comparisons between MoCap data based methods and video pseudo annotation based methods (L34-38 of the main submission). We want to express that the model capacity is limited due to the limited dataset scale. Sorry for the misunderstanding. We have revised the writing in the revised version.
>
> **2)** As shown in previous studies [c, d], the co-speech gesture motions and styles vary a lot for different speakers, which is termed as “individual speaking style” [c]. Therefore, it is suitable to train a separate co-speech gesture generation model for each person following the experiment settings of baselines [a, c, d]. However, we explore a more challenging task of co-speech gesture image generation in a unified framework without structural prior and achieve superior performance. Even for a single-person subset, it is non-trivial to animate non-rigid human body in image space by speech audio, especially with the interference of complex background scenes.
>
> **3)** It is difficult to achieve a person-agnostic co-speech gesture **image** generation model with **currently available datasets**. In particular, the commonly used datasets are TED Gesture [h] and PATS [c, d]. TED Gesture is based on TED Talk videos, while PATS contains 25 speakers of talk shows, lectures, etc. Due to the frequent camera movements and viewpoint shift in TED videos, there lacks clear co-speech gesture clips for **image** generation. Hence we narrow down the experiments to PATS dataset in this work. A dataset with high-quality co-speech gesture image frames of multiple speakers is needed to learn a model of novel person generalization ability. We will strive for this in future work.
>
> **4)** We verify the potential generalization ability of our approach in two aspects:
>
> * We could animate the same speaker’s different appearances with speech audio (as shown in the codebook analysis part of demo video, we could animate Oliver’s different appearances), while previous studies [a, c] that resort to off-the-shelf pose2img generator only support a single appearance.
>
> * We additionally implement the experiments of animating with unseen audio from a different person. The evaluated results are reported below. It shows that the model’s performance is still effective with the unseen audio input. With the proposed vector quantize design, each codebook entry defines a reasonable co-speech gesture pattern. In contrast to directly mapping to the continuous coordinate space, such technical design guarantees a valid gesture even when generalizing to the unseen audio from a different person. The results are included in Section E of the supplemental document.
>
> |Methods|FGD$\downarrow$|BC$\uparrow$|Diversity$\uparrow$|
> |-|-|-|-|
> |ANGIE (Novel Audio)|1.46|0.69|48.5|
> |ANGIE (Ours)|1.35|0.72|49.4|

---

> ### Author Response · Authors · 2022-08-02
> **Response to Reviewer S12n [Part 1/3]**
>
> We sincerely thank the reviewer for your insightful and constructive feedbacks. We have polished the paper, added the experiments and made the clarifications in the revised version.
>
> The technical contributions and novelty of this work are highlighted in the General Response. Please kindly refer to it for details. Note that the following polishments have been made according to your advice:
>
> * We have corrected the misunderstanding of “generalize ability” by ”capacity” in L35, L82 of the main submission.
> * We have highlighted “position-irrelevant” denotes “image location invariant” in L194 of the main submission.
> * We have included the analysis on the model’s generalization ability in Section E of the supplemental document.
> * We have added the results for unseen audio from a different person in Section E of the supplemental document.
> * We have included the analysis on additional text information input in Section H of the supplemental document.
> * We have included the discussion of model’s ability in conversation in Section D of the supplemental document.
>
> Thanks again for your very constructive comments, which have helped us improve the quality of the paper significantly! Below we would like to first address your common concerns, then provide point-to-point responses to all the raised questions.
>
> > **Common Q1: "Why not use the language model or text (semantic) information in this work, so that some semantics relevant gestures like iconics, metaphorics, and deictics are unlikely to be generated. Only the beat gestures could be generated."**
>
> **Common A1:** Thank you for the precious advice! We give our responses below:
>
> **1)** We would like to clarify that our main focus and technical contributions are how to animate the co-speech gesture in the image domain, but not on the influence of input modality. We choose the audio-driven setting mainly for two reasons:
>
> * We generally follow the problem setting of baselines [a, c, d], where all the compared methods take only the speech audio as input modality for fair comparison.
>
> * In order to use the text information, we have to pre-process the raw transcripts to be temporally aligned with audio using tools like Gentle. To prevent the potential alignment inaccuracy in the pre-processing step, and to simplify the problem setting for better focusing on audio-driven co-speech gesture image animation, we do not involve the text input in this work.
>
> **2)** As proved in Automatic Speech Recognition (ASR) [e, f] and recent co-speech gesture studies [g], the speech audio actually contains some high-level semantic information. Such implicit semantic information in the speech audio could guide the model to capture some specific co-speech gesture patterns like metaphorics [g]. Besides, as shown in the Kubinec chemistry lecture setting, our model manages to synthesize deictic gestures of pointing to the screen by learning such implicit audio-gesture correlations (the reasons and analysis on why we could learn this are elaborated in **Common A2**). We would like to respectfully claim that the generated gestures of our model are not only limited to the beat gestures, but some semantic gestures could also be synthesized.
>
> **3)** We agree with the reviewer that language model/text information contains rich semantics, which are beneficial to the learning of semantics relevant gestures like iconics, metaphorics, and deictics. Previous study [h] has also verified the influence of each input modality on co-speech gesture, including audio, text and speaker identity, etc. Therefore, we additionally complement an experiment of using text feature: we encode the transcripts by TextTCN [i] and further concatenate with audio features. The combined audio text features are fed into Co-Speech GPT with Motion Refinement network to predict the quantized code as well as motion residuals. The results are reported below, which suggest that the text feature could indeed facilitate better co-speech gesture generation. We have included the experiment results in Section H of the supplemental document.
>
> |Methods|FGD$\downarrow$|BC$\uparrow$|Diversity$\uparrow$|
> |-|-|-|-|
> |ANGIE (Extra Text Concat)|1.30|0.73|49.8|
> |ANGIE (Ours)|1.35|0.72|49.4|
>
> **4)** As an early attempt to explore audio-driven co-speech image generation, this work could serve as a baseline for further studies in the research community. However, how to effectively fuse the multiple modality information (including audio and text) and better map to the implicit motion representation remains an open problem. We will explore this in future work.

---

### Official Review · Reviewer_FuNA · 2022-07-11

**Rating:** 7
**Confidence:** 3
**Soundness:** 4 excellent
**Presentation:** 2 fair
**Contribution:** 3 good

**Summary:**

The paper presents ANGIE, an approach for audio-driven image animation with specific focus on upper body gestures. ANGIE leverages an implicit motion model (MRAA) as the intermediate motion representation instead of operating on a landmark-based skeleton representation which allows it to work directly in the image domain. The problem is divided into 3 parts: a motion representation learning module, a motion prediction module, and a motion refinement module. First, a VQVAE is trained to discretize the motion into a fixed set of codes. This model learns a position-independent codebook of co-speech gestures. Next, a Co-Speech Gesture GPT learns to map audio onset features to the discrete codes of motion extracted from training videos. This GPT network is also trained with a residual loss which focuses on refining the motion predicted from the discrete codes.

The authors compare their approach to previous state-of-the-art audio-driven gesture generation networks and evaluate using objective metrics and subjective user studies. Overall, the proposed method outperforms the baselines. One of the baselines: HA2G tends to have better or similar beat consistency than the proposed method. The authors also perform ablations studies to compare different vector quantization strategies and confirming the benefit of the motion refinement step. The authors also analyze the quantized motion representations.

**Questions:**

- L140: Shouldn't the output frames be from I^hat_(2:N) if I_1 is given? If that is the case, perhaps the authors can just say I_0 is given and then the rest of the paper need not be changed.
- How is the onset feature strength computed? What is the window size or hop size for spectral flux computation? I don't understand what the 426 dimensions are in the onset feature. Onset strength should be a one dimensional feature per frame. Do you mean to say that the training videos are a fixed length and have 496 audio frames? It does seem like the videos are fixed length (96 frames @ 25 fps) but then you also need to mention some technical specifications: sample rate of the audio.
- Similarly, how are the mfcc features computed? Again, I don't understand the dimensions of the MFCCs. Are you using 12 mfccs or 28 mfccs. In any case, if the onset strength length is 426, why is the temporal dimension of the mfccs so small? And you mention that they are computed with a window size of 10 ms. What about hop size? What about fft block size?
- Does the network also infer mouth movement? It would seem that way since there is no special treatment of head/face from the body gestures, but then I am really surprised by the quality of the mouth motion in the sample videos shared. Why have the authors not also evaluated using audio-driven face animation metrics?


**Limitations:**

The authors adequately discuss the limitations of their work along with ethics considerations of such algorithms.

**Strengths And Weaknesses:**

Strengths:

- This paper is the first to animate face and torso images without using a skeleton representation of the images and demonstrates improved performance over those approaches.
- The method is another interesting application of VQ-VAE followed by a prediction model. The novelty also lies in the design of the VQVAE. Instead of naively learning a codebook for the motion features of MRAA, the authors transform their data to address constraints in their problem, such as learning a codebook from the lower triangular decomposition of their covariance matrix and using relative motion parameters instead of absolute parameters. Their decisions are also ablated in their evaluation showing the benefits of these transformations.

Weaknesses:

There are some minor issues with the paper in terms of details.
- The subjective evaluation involves some kind of mean opinion score (MOS). The authors do not add 95% confidence intervals or a standard deviation to their scores. This is important to see the spread in the scores given to each system.
- Some details regarding audio feature extraction are missing in the paper. Please refer to the questions section.

---

> ### Author Response · Authors · 2022-08-02
> **Response to Reviewer FuNA**
>
> We sincerely thank the reviewer for your insightful comments and recognitions to this work, especially for acknowledging that our approach is technically sound with novel vector quantization design. We have corrected the typo and made the clarifications in the revised version.
>
> Note that the following polishments have been made according to your advice:
>
> * The standard deviation of user study result is included in Section F of the supplemental document.
> * The variable index typo is corrected in Section 3.1 of the main submission.
> * The details on audio feature extraction are included in Section G of the supplemental document.
>
> Thanks again for your very constructive comments, which have helped us improve the quality of the paper significantly! Below we would like to provide point-to-point responses to all the raised questions:
>
> > **Q1: "The subjective evaluation involves mean opinion score (MOS). The authors do not add 95% confidence intervals or a standard deviation to their scores to see the spread in the scores given to each system."**
>
> **A1:** Many thanks for your precious advice! The standard deviation of each user study score is reported below, which is included in Section F of the supplemental document. Besides, in the initial submission we have measured the participants’ scoring disagreement with Fleiss’s-Kappa statistic in L301 of the main submission. The value shows that the agreement among scorers is highly consistent.
>
> |Methods|Ground Truth|S2G|HA2G|SDT|TriCon|ANGIE (Ours)|
> |-|-|-|-|-|-|-|
> |Realness|0.492|0.413|0.293|0.206|0.385|0.312|
> |Synchrony|0.480|0.574|0.214|0.265|0.350|0.345|
> |Diversity|0.252|0.629|0.471|0.313|0.221|0.287|
>
> > **Q2: "L140: Shouldn't the output frames be from I^hat_(2:N) if I_1 is given? If that is the case, perhaps the authors can just say I_0 is given and then the rest of the paper need not be changed."**
>
> **A2:** Thank you for pointing out this typo! We have corrected the variable indexes according to your advice in Section 3.1 of the main submission. Please kindly check the highlighted changes of blue text color in the formula and problem definition part.
>
> > **Q3: "How is the onset feature strength computed? What is the window size or hop size for spectral flux computation? What the 426 dimensions are in the onset feature?"**
>
> **A3:** The audio onset strength feature of T frames is of shape (T, 426), where T is the temporal dimension (frame number) and 426 is the feature channel dimension. It is the concatenation of constant-Q chromagram, tempogram, onset beat, onset tempo and onset strength. Most features are derived from the audio onset strength/envelope and the channel dimension is summed up to 426. We utilize the librosa onset functions to extract the features, including “librosa.onset.onset_strength”, “librosa.feature.tempogram” and “librosa.beat.beat_track”, etc. The audio sample rate is 16000, the time lag for computing differences is 1, the hop length is 512 and the window length is 384.
>
> > **Q4: Similarly, how are the mfcc features computed? Again, I don't understand the dimensions of the MFCCs. Are you using 12 mfccs or 28 mfccs. In any case, if the onset strength length is 426, why is the temporal dimension of the mfccs so small? And you mention that they are computed with a window size of 10 ms. What about hop size? What about fft block size?"**
>
> **A4:** The original audio mfcc feature was calculated as 12 mfccs which has the dimension of (T’, 12) and T’ is the original audio frame number. In our implementation, we use a 28-dim sliding window to further unfold the mfcc feature into a final shape of (T, 28, 12), where T denotes the final temporal dimension (video frame number), 28 denotes the size of sliding window and 12 is the mfcc feature dimension. MFCC feature is extracted with a sample rate of 16000, window length of 25 ms, window step of 10 ms, cepstrum number of 13, filters number of 26 and FFT block size of 512. In the motion refinement module, we use the a certain frame’s mfcc feature of shape (28, 12) and forward a series of convolution and linear layers to extract the per-frame audio feature of dimension 128.
>
> > **Q5: Does the network also infer mouth movement? It would seem that way since there is no special treatment of head/face from the body gestures. Why have the authors not also evaluated using audio-driven face animation metrics?"**
>
> **A5:** Our main focus in this work is the upper body co-speech gesture. We follow previous studies [a, b] that post-process the facial movement merely for demo visualization. Please kindly refer to L105-117 of the supplemental document for more details, analysis and discussions.
>
> ****
>
> [a] - Ginosar et al. "Learning Individual Styles of Conversational Gesture."
>
> [b] - Qian et al. "Speech Drives Templates: Co-Speech Gesture Synthesis with Learned Templates."
>
> ****
>
> Please don’t hesitate to let us know if there are any additional clarifications or experiments that we can offer!

---

> ### Author Response · Authors · 2022-08-05
> **Additional Response to Reviewer FuNA**
>
> Dear Reviewer FuNA:
>
> We sincerely thank you again for your great efforts in reviewing this paper, especially for the precious advice that has helped us improve the quality of this paper significantly!
>
> We have included the user study standard deviation, fixed the variable index typo and elaborated the audio feature extraction details in the revised version. Please don't hesitate to let us know if there are further clarifications or experiments that we could offer!
>
> Best,
>
> Paper 365 Authors.

---

> > ### Comment · Reviewer_FuNA · 2022-08-08
> > **Response to authors**
> >
> > I thank the authors for responding to my review with various changes and additions to the paper.
> >
> > I am mostly satisfied with the responses. I have one follow-up question though about the onset features used. Is there any motivation behind using the constant-Q chromagram? All the other features make sense for onset or rhythm information, but the chromagram is typically used in music information retrieval to track harmonic changes. If you are expecting the change in energy in this feature to be informative to the network, then that is already captured in the spectral flux-based onset strength that you extracted from librosa.

---

> > > ### Author Response · Authors · 2022-08-09
> > > **Thanks for your additional comments!**
> > >
> > > We sincerely thank the reviewer for the additional feedbacks. We use the constant-Q chromagram mainly for two reasons: **1)** As mentioned in L212 of the main submission, previous studies [a, b] suggest that the onset strength information is more suitable for cross-modal pattern learning. Therefore, we generally follow [a] to use the constant-Q chromagram as one of the input features to predict the quantized motion pattern codes. **2)** Since the chromagram could reflect the harmonic and energy changes, we use it as supplemental information to the onset features in a more explicit manner. We agree with the reviewer that some chromagram information is already captured in the spectral flux-based onset strength. We will include the ablation experiments on the audio feature choice in the final version.
> > >
> > > We are delighted to hear that the reviewer is mostly satisfied with our responses! Many thanks again for your very constructive comments, which have helped us improve the quality of the paper significantly. Please don't hesitate to let us know if there are further clarifications that we could offer!
> > >
> > > ****
> > >
> > > [a] - Li et al., "Bailando: 3D Dance Generation by Actor-Critic GPT with Choreographic Memory."
> > >
> > > [b] - Tang et al., "Dance with Melody: An LSTM-Autoencoder Approach to Music-Oriented Dance Synthesis."

---

> ### Comment · Area_Chair_tPJc · 2022-08-08
> **Reviewer FuNA: please respond to the authors**
>
> The authors have provided standard deviations of the MOS scores and explicated their feature processing. Please read their response and reply.
>
> Thanks

---

### Official Review · Reviewer_SBef · 2022-07-11

**Rating:** 6
**Confidence:** 4
**Soundness:** 3 good
**Presentation:** 3 good
**Contribution:** 3 good

**Summary:**

This paper looks into the topic of generating gesture videos from given speech and images. The proposed approach learns the representations of gesture motions as a codebook, and learns the mapping from the speech to the codebook. With the learned representation and the speech-to-codebook mapping, the approach can animate the given image with gestures based on the given speech. The speech representation for the image is based on previous work “Motion Representations for Articulated Animation”. Since the codebook provides quantized representation and can miss some details, the approach includes an additional residual learning approach to further construct the image details from speech. User studies showed improvement compared to previous work. Ablation studies are provided to show the effectiveness of the proposed techniques.

**Questions:**

How well does the approach generalize to speakers that are not in the training data?

**Limitations:**

The authors have properly discussed the limitations.

**Strengths And Weaknesses:**

Strength:
The approach generates good quality gesture videos based on the given image, and showed improvement compared to previous work.

Ablation studies demonstrated the effectiveness of the proposed techniques.


Weaknesses:
The idea of learning representations and learning the mapping from utterances to the representations are not new, and the motion representations used in the work is proposed in the previous work. The main novelty is applying the idea on gesture generations.

It is not clear how the approach performs on speakers that are not in the training data.

---

> ### Author Response · Authors · 2022-08-02
> **Response to Reviewer SBef**
>
> We sincerely thank the reviewer for your insightful comments and recognitions to this work, especially for acknowledging that our approach is technically effective with superior performance. We have polished the paper, added the experiments and clarified the below points in the revised version.
>
> > **Q1: "Learning the representation and mapping from utterances to the representations are not new. The main novelty is applying the idea on gesture generations."**
>
> **A1: 1)** We would like to clarify that our main novelty lies in the vector quantize design to extract the common motion pattern and residual refinement to complement subtle movement details (as recognized by Reviewer FuNA, WZ3S), while the motion representations are not the key focus of this paper. Therefore, we follow MRAA and introduce them merely as a preliminary section for self-contained contents.
>
> **2)** As specified in L172-174 of the main submission, our mapping design from utterances to representations is **actually different from previous studies** in two-folds:
>
> * Previous studies directly map utterances to pose coordinates in a continuous space, which is a _harder regression_ problem. On the contrary, we ease the problem by predicting a category of quantized codebook (i.e., codebook entry), which is an _easier classification_ problem. We thus alleviate the cross-modal audio-to-gesture learning difficulty.
>
> * With the quantized motion code sequences, we could use powerful attention-based Transformer for better mapping learning (as recognized by Reviewer WZ3S).
>
> **3)** As elaborated in the General Response, we provide a solution on how to deal with the constrained vector quantization problem and how to complement sequential results with missing details. Such novelty could potentially benefit relevant researches like constrained vector quantization problem and video generation, which is not limited to co-speech gesture generation.
>
> > **Q2: "How well does the approach generalize to speakers that are not in the training data?"**
>
> **A2: 1)** As shown in previous studies [a, b], the co-speech gesture motions and styles vary a lot for different speakers, which is termed as “individual speaking style” [b]. Therefore, it is suitable to train a separate model for each person following the baseline’ experiment settings [a, b, c]. However, we explore a more challenging task of co-speech gesture image generation in a unified framework without structural prior and achieve superior performance. Even in a single-person subset, it is non-trivial to animate non-rigid human body in image space by speech audio, especially with the interference of complex background scenes.
>
> **2)** It is hard to generalize to speakers that are not in the training data with **currently available co-speech gesture image datasets**. In particular, the commonly used datasets are TED Gesture [d] and PATS [a, b]. TED Gesture is based on TED Talk videos, while PATS contains 25 speakers of talk shows, lectures, etc. Due to the frequent camera movements and viewpoint shift in TED videos, there lacks clear co-speech gesture clips for **image** generation. Hence we narrow down the experiments to PATS dataset in this work. A dataset with high-quality co-speech gesture image frames of multiple speakers is needed to learn a model of unseen person generalization ability. We will strive for this in future work.
>
> **3)** We verify the potential generalization ability of our approach in two aspects:
>
> * We could animate the same speaker’s different appearances with speech audio (as shown in the codebook analysis part of demo video, we could animate Oliver’s different appearances), while previous studies [b, c] that resort to off-the-shelf pose2img generator only support a single appearance.
>
> * We additionally implement the experiments of animating with unseen audio from a different person. The evaluated results are reported below. It shows that the model’s performance is still effective with the unseen audio input. With the proposed vector quantize design, each codebook entry defines a reasonable co-speech gesture pattern. In contrast to directly mapping to the continuous coordinate space, such technical design guarantees a valid gesture even when generalizing to the unseen audio from a different person.
>
> |Methods|FGD$\downarrow$|BC$\uparrow$|Diversity$\uparrow$|
> |-|-|-|-|
> |ANGIE (Novel Audio)|1.46|0.69|48.5|
> |ANGIE (Ours)|1.35|0.72|49.4|
>
> ****
>
> [a] - Ahuja et al. "Style Transfer for Co-Speech Gesture Animation: A Multi-Speaker Conditional-Mixture Approach."
>
> [b] - Ginosar et al. "Learning Individual Styles of Conversational Gesture."
>
> [c] - Qian et al. "Speech Drives Templates: Co-Speech Gesture Synthesis with Learned Templates."
>
> [d] - Yoon et al. "Speech Gesture Generation from the Trimodal Context of Text, Audio, and Speaker Identity."
>
> ****
>
> Please don’t hesitate to let us know if there are any additional clarifications or experiments that we can offer!

---

> ### Author Response · Authors · 2022-08-05
> **Additional Response to Reviewer SBef**
>
> Dear Reviewer SBef:
>
> We sincerely thank you again for your great efforts in reviewing this paper, especially for the precious advice that has helped us improve the quality of this paper significantly!
>
> We have clarified the novelty of this work, included the discussions on model's generalization ability and implemented additional experiments in the revised version. Please don't hesitate to let us know if there are further clarifications or experiments that we could offer!
>
> Best,
>
> Paper 365 Authors.

---

> ### Comment · Area_Chair_tPJc · 2022-08-08
> **Reviewer SBef: please respond to the authors**
>
> The authors have directly addressed your question: "How well does the approach generalize to speakers that are not in the training data?"
>
> Please read their response and reply to it.
>
> Thanks

---

### Author Response · Authors · 2022-08-02
**General Response**

We sincerely thank all the reviewers for your constructive feedbacks and recognitions to this work, especially for acknowledging that **the novel vector quantize design could benefit further research** (Reviewer FuNA, WZ3S), **the motion refinement module is novel** (Reviewer WZ3S), **the performance is superior** (all reviewers), and **the ablation study is thorough** (Reviewer SBef, FuNA, WZ3S). We have polished the paper, added the experiments, and made the clarifications in the revised version.

We would like to re-emphasize the novelty and technical contributions of this work:

* To the best of our knowledge, we are one of the earliest attempts to explore such a challenging setting of generating co-speech gesture images in a unified framework without structural prior annotation. Actually, it is non-trivial to animate non-rigid human body in image space by speech audio, especially with the interference of complex background scenes. In spite of this, a novel framework is proposed with superior performance than baselines. We sincerely hope that our contributions could be appreciated.

* We design two novel modules named VQ-Motion Extractor and Co-Speech GPT with Motion Refinement. **1)** Instead of naively applying VQ-VAE to motion representations, we design a cholesky decomposition strategy to solve the constrained vector quantization problem (i.e., guarantee that the reconstructed covariance matrix is symmetric positive definite). **2)** Then, we improve the quantization scheme to encode the relative motion representation that is position (absolute location) irrelevant. **3)** The motion refinement module is further devised to complement the subtle motion details. The effectiveness of all modules is verified by extensive experiments.

* As an early attempt to explore audio-driven co-speech image generation, this work could pave way for further studies in the audio-visual generation community. Besides, our approach gives an idea on how to deal with the constraints in vector quantization and how to complement sequential results with missing details. We hope this paper could provide insights for relevant domains like constrained vector quantization problem and video generation tasks.

****

We have revised our manuscript to include the following changes according to all the reviewers’ insightful comments. Note that all the polishments on the main submission and supplemental document are highlighted with **blue** text color for better visualization.

* We have polished to highlight the potential impact of our work on relevant research domains in Section “Related Work” of the main submission.

* We have corrected several typos/misunderstandings in the main submission, which includes: change the wording from “generalization ability” to ”capacity” (L35, L82); correct the variable index typo (Section 3.1); highlight that the “position-irrelevant” denotes “image location invariant” (L194); eliminate the Wikipedia article reference (L301) and elaborate the concept of Fleiss's Kappa statistic in the supplemental document (Section J).

* We have included the analysis on the model’s generalization ability in Section E of the supplemental document.

* We have added the experimental results for unseen audio from a different person in Section E of the supplemental document.

* We have included the user study score standard deviation in Section F of the supplemental document.

* We have included the audio feature extraction details in Section G of the supplemental document.

* We have included the analysis on additional input modality of text information in Section H of the supplemental document.

* We have included the discussion of model’s potential ability in general conversational setting in Section D of the supplemental document.

* We have included the motion estimator training details in Section C of the supplemental document.

Please don't hesitate to let us know of any additional comments on the manuscript or the changes.

---

### Meta-Review · Area_Chair_tPJc · 2022-08-24

**Recommendation:** Accept
**Confidence:** Certain

**Metareview:**

This paper enjoyed a reasonable interaction between the authors and the reviewers, with the authors addressing the reviewers' concerns about the novelty of the proposed method, its specificity to the "talking head" scenario, the fact that the model is used in a speaker-dependent fashion, and some concerns about specific details in the writing. Three of the four reviewers responded to the authors during the discussion period, and the fourth reviewer acknowledged having read the rebuttal during the discussion between the AC and reviewers.

In the end, all reviewers recommend acceptance of the paper, citing the good performance of the model, the novelty of co-speech gesture generation in the image domain, and the nice design of the model (specifically, the VQ motion plus residual structure).


**Award:**

No

---

### Decision · Program_Chairs · 2022-09-14

Accept